# The industrial anaerobe *Clostridium acetobutylicum* uses polyketides to regulate cellular differentiation

Nicolaus A. Herman[1], Seong Jong Kim[1], Jeffrey S. Li[1], Wenlong Cai[1], Hiroyuki Koshino[2] & Wenjun Zhang[1,3]

Polyketides are an important class of bioactive small molecules valued not only for their diverse therapeutic applications, but also for their role in controlling interesting biological phenotypes in their producing organisms. While numerous polyketides are known to be derived from aerobic organisms, only a single family of polyketides has been identified from anaerobic organisms. Here we uncover a family of polyketides native to the anaerobic bacterium *Clostridium acetobutylicum*, an organism well-known for its historical use as an industrial producer of the organic solvents acetone, butanol, and ethanol. Through mutational analysis and chemical complementation assays, we demonstrate that these polyketides act as chemical triggers of sporulation and granulose accumulation in this strain. This study represents a significant addition to the body of work demonstrating the existence and importance of polyketides in anaerobes, and showcases a strategy of manipulating the secondary metabolism of an organism to improve traits relevant for industrial applications.

[1] Department of Chemical and Biomolecular Engineering, University of California-Berkeley, Berkeley, CA 94720, USA. [2] RIKEN Center for Sustainable Resource Science, Wako, Saitama 351-0198, Japan. [3] Chan Zuckerberg Biohub, San Francisco, CA 94158, USA. Correspondence and requests for materials should be addressed to W.Z. (email: wjzhang@berkeley.edu)

Polyketides are one of the most important classes of natural products given their wide range of applications in medicine and agriculture[1,2]. Encompassing several different chemical classes such as macrolides, polyenes, and aromatics, polyketides are employed for clinical use as antibiotics, anti-cancer agents, immunosuppressants, and even cholesterol-lowering drugs[3]. In addition to their role as therapeutic agents, many of these compounds are used by their producing organisms to access information about both the intracellular physiological status and extracellular environment, and control complex cellular processes such as morphological differentiation, virulence, stress response, and additional secondary metabolite production[4]. As virtually all known polyketides are derived from aerobic organisms (such as bacteria, fungi, and plants), there has been a long-standing assumption that anaerobic organisms are unable to produce these compounds[5]. However, recent genomic analysis has challenged this view by revealing that polyketide biosynthetic genes are widespread among anaerobic bacteria, and in particular, members from the genus *Clostridium*[6]. This diverse genus, comprised of anaerobic endospore-forming Gram-positive firmicutes, includes several notorious human pathogens as well as non-pathogenic species useful for industrial biotechnology[7]. To date, only one family of polyketides, the clostrubins, has been identified from *Clostridium*, representing the only known polyketides from the anaerobic world[8]. Clostrubins are pentacyclic polyphenolic polyketides biosynthesized by type II polyketide synthases (PKSs) in *Clostridium beijerinckii*[8] and *Clostridium puniceum*[9], and display potent antibiotic activity against various pathogenic bacteria. Additionally, these aromatic polyketides have been shown to enable the plant pathogen *C. puniceum* to survive in an oxygen-rich environment. The discovery of clostrubins provided the first experimental evidence that anaerobes are capable of producing bioactive polyketides, and motivated further studies on anaerobes to reveal additional polyketide metabolites with novel structures and interesting biological functions.

*Clostridium acetobutylicum* is an organism historically used for industrial-scale production of the organic solvents acetone, *n*-butanol, and ethanol (ABE) through a process known as ABE fermentation[10]. Batch ABE fermentation by *C. acetobutylicum* ATCC 824 (the model ABE producer) is characterized by two distinctive phases, an acid production phase (acidogenesis) and a solvent production phase (solventogenesis)[11]. During exponential growth, short-chain fatty acids (acetate and butyrate) are produced and accumulate in the media, causing a drop in the culture pH. As the culture approaches stationary phase, the previously formed acids are re-assimilated, the culture pH rises, and solvent production is initiated. The metabolic switch from acidogenesis to solventogenesis coincides with the initiation of the complex development program of sporulation[12]. As part of these processes, a starch-like carbohydrate known as granulose is produced and accumulates in swollen, phase-bright clostridial forms, within which endospores develop[13]. Further morphological development yields free spores, heat- and chemical-resistant cell types that do not contribute to solvent production.

Although ABE fermentation of *C. acetobutylicum* has been extensively studied, chemical signals responsible for triggering solventogenesis and/or sporulation have not been elucidated, and no secondary metabolite that plays a role in regulating these processes has been reported from this organism. Motivated by a recent transcriptomic analysis of *C. acetobutylicum*[14], which showed that the expression of a type I modular PKS gene was significantly upregulated during early stationary phase (~40-fold higher transcription level compared to mid-exponential phase), we postulated that the corresponding polyketide product might regulate one or more of the key fermentation phenotypes associated with solventogenesis and sporulation. Here we report the discovery of a family of polyketide metabolites using comparative, untargeted metabolomics followed by large-scale compound purification and molecular structure elucidation. Using a reverse genetics approach and chemical complementation, we further

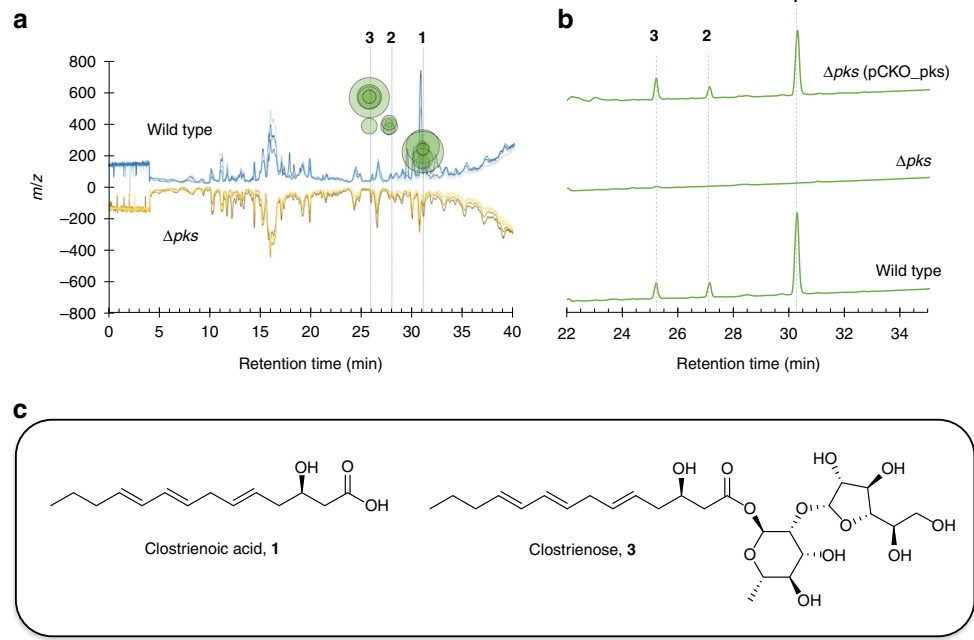

**Fig. 1** Identification of polyketides from wild-type extracts. **a** Results of XCMS analysis of wild-type *C. acetobutylicum* and Δ*pks* extracts taken from early stationary phase fermentation cultures (biological quadruplicates). Blue (upper) chromatograms depict MS traces of quadruplicate wild-type extracts, while orange/brown (lower) chromatograms depict MS traces of quadruplicate Δ*pks* extracts. Green circles represent MS peaks unique to wild-type extracts, with corresponding *m/z* values indicated by the *y*-axis. No significant MS peaks were identified, which were unique to Δ*pks*. The three peaks unique to wild type are identified as **1**, **2**, and **3**. **b** UV–Vis traces (240 nm) of extracts from wild type, Δ*pks*, and Δ*pks* genetic complementation strain (Δ*pks* (pCKO_pks)). **c** Elucidated structures of **1** and **3** based on NMR characterization

show that deletion of the *pks* gene results in increased batch butanol production, and that the newly discovered polyketides are important in stimulating sporulation and granulose accumulation in *C. acetobutylicum*.

## Results

**Analysis and inactivation of *pks* locus in *C. acetobutylicum*.** All sequenced *C. acetobutylicum* strains possess one *pks* gene (*ca_c3355* in *C. acetobutylicum* ATCC 834) that encodes a type I single-module PKS with a predicted catalytic domain organization of ketosynthase, acyltransferase, dehydratase, ketoreductase, and acyl carrier protein (KS-AT-DH-KR-ACP) (Supplementary Fig. 1a). Homologous PKSs with the same domain structure could be identified, but the genes for these homologous PKSs are typically part of much larger biosynthetic gene clusters that encode additional PKSs and/or modification enzymes. *ca_c3355* appears to be an orphan gene with no other biosynthetic enzyme encoded either upstream or downstream, although several transcriptional regulators and efflux pumps are encoded in the vicinity (Supplementary Fig. 1b). While we were able to predict that malonyl-CoA was the likely substrate recognized by the AT domain (Supplementary Fig. 1a), it was difficult to predict the identity of polyketides, which might be synthesized by this PKS, or if any chemical modifications of the nascent polyketide intermediate would be made by proteins encoded elsewhere on the genome. To determine the identity of any PKS-associated metabolites and probe the broader function of the *pks* locus, we performed a targeted in-frame deletion of the *pks* gene (*ca_c3355*) using an allelic exchange method developed for *C. acetobutylicum*[15]. The resulting mutant (Δ*pks*) was confirmed by PCR analysis (Supplementary Fig. 2).

**Discovery of polyketides through comparative metabolomics.** Quadruplicate batch fermentations with wild-type *C. acetobutylicum* and the mutant Δ*pks* were performed and harvested at early stationary phase. Organic extracts from combined supernatants and cell pellets were obtained and analyzed via liquid chromatography-high-resolution mass spectroscopy (LC-HRMS). Following untargeted metabolomic comparisons of the two strains using XCMS[16], three major species with molecular formulas $C_{14}H_{22}O_3$ (**1**, calculated for $C_{14}H_{21}O_3^-$: 237.1496; found: 237.1496), $C_{20}H_{32}O_7$ (**2**, calculated for $C_{20}H_{31}O_7^-$: 383.2075; found: 383.2080), and $C_{26}H_{42}O_{12}$ (**3**, calculated for $C_{26}H_{41}O_{12}^-$: 545.2604; found: 545.2600) were found to be present in wild-type culture extracts and completely absent in extracts of Δ*pks* (Fig. 1a, b; Supplementary Figs. 3–5). A majority of **1–3** was found in the culture medium rather than cell pellets, suggesting that they were secreted into the culture medium upon production. **1–3** were UV active and featured identical UV absorption spectra (Supplementary Figs. 3–5), indicating that they share the same chromophore and likely the same biosynthetic origin. Furthermore, when the *pks* gene was introduced back into the Δ*pks* mutant, the genetic complementation strain was found to have resumed production of **1–3**, demonstrating a direct relationship between the *pks* gene and the production of **1–3** (Fig. 1b).

**Isolation and structure elucidation of polyketides.** To isolate compounds **1–3**, we prepared 34 L of wild-type *C. acetobutylicum* culture broth. The culture was extracted with ethyl acetate and chromatographed on a silica gel column, followed by further purification via multiple rounds of HPLC using reverse-phase C18 columns (Supplementary Methods). These purification steps yielded pure compound **1** (1.1 mg) and **3** (0.9 mg). NMR spectra, including $^1H$, $^{13}C$, DQF-COSY, TOCSY, HSQC, and HMBC

spectra, were obtained for compound **1** (Supplementary Table 1; Supplementary Fig. 6; Supplementary Methods). The proton signals at δ 5.43, 5.56, and 6.00 indicated a considerable overlap in the aliphatic double bond region, and their configuration (all *E* stereochemistry) was further resolved using the high-resolution HSQC spectrum without $^{13}C$ decoupling. These assignments of geometric stereochemistry were also supported from $^{13}C$ chemical shift values of allylic methylenes[17]. The carbon signals at δ 172.97 and 67.25, together with related HMBC and TOCSY correlations, indicated the presence of a carboxylic acid and a secondary alcohol. The absolute stereochemistry of **1** was determined to be in the *R* configuration by measuring the specific rotation of its fully reduced product, 3-hydroxytetradecanoic acid (3-HTA). In particular, **1** was reduced to 3-HTA through hydrogenation over a palladium catalyst, and the specific rotation of the product ($[\alpha]_D^{20}$−15.7° (*c* 1, CHCl$_3$)) was consistent with the reported value of optically pure (*R*)-3-HTA ($[\alpha]_D^{20}$−16.2° (*c* 1, CHCl$_3$))[18] (Supplementary Methods). From these data, we elucidated the molecular structure of **1** to be a modified tetradecenoic acid, which we termed clostrienoic acid (Fig. 1c). The predicted molecular formulas, MS/MS analysis, and identical UV absorption spectra suggested that **2** and **3** were related to **1**, likely containing additional monosaccharide and disaccharide moieties, respectively (Supplementary Figs. 3–5). Further 1D and 2D NMR spectroscopic analysis revealed that a disaccharide, α-D-galactofuranosyl(1→2)-α-L-rhamnopyranoside, is connected to the backbone of clostrienoic acid through an ester linkage in **3**, which we termed clostrienose (Fig. 1c; Supplementary Table 2; Supplementary Fig. 7; Supplementary Methods). It is notable that the same disaccharide sequence has been found in several polysaccharides produced by microorganisms, and the relative configuration of each monosaccharide substituent in **3** was further confirmed by comparison to previously published NMR spectra[19–22]. An ester linkage between a sugar moiety and acid is rare, and only a few naturally occurring glycosyl esters have been previously reported, such as the β-D-glycosyl ester of 5-isoprenylindole-3-carboxylate isolated from *Streptomyces* sp. RM-5-8[23], a microbial modification product of A58365A from *Streptomyces chromofuscus* NRRL 15098[24], and a glycosyl ester of 3,4-seco-triterpene[25]. The minor compound **2** was indicated to contain one monosaccharide substituent (α-L-rhamnopyranoside) based on HRMS/MS analysis, but the titer was too low to be further confirmed by NMR spectroscopic analysis (Supplementary Fig. 4).

**In vitro biochemical analysis of PKS.** Based on the structures of **1** and **3**, we hypothesized that the single-module PKS could use malonyl-CoA substrates and function iteratively to yield a heptaketide intermediate. Further modifications by putative auxiliary enzymes such as enoyl reductase, isomerase/desaturase, and thioesterase would then form the clostrienoic core (**1**). To confirm that *ca_c3355* encodes a functional iterative type I PKS, the 203 kDa megasynthase was overexpressed and purified from *Escherichia coli* strain BAP1[26]. After incubating the purified PKS with $^{14}C$-labeled malonyl-CoA (generated in situ from 2-$^{14}C$-malonic acid and CoA by malonyl-CoA synthase [MatB][27]), the $^{14}C$-labeled covalent acyl-*S*-thiolation intermediate was detected by SDS–PAGE autoradiography (Supplementary Fig. 8). Further in vitro reconstitution of the activity of PKS with malonyl-CoA showed a dominant product, the triketide lactone **4**, which is a typical shunt product after spontaneous cyclic off-loading of the unreduced triketide. Addition of NADPH to this reaction enabled function of the KR domain (and the subsequent DH domain), leading to the production of the known shunt products tetraketide pyrone **5** and pentaketide pyrone **6** (Supplementary Figs. 8,

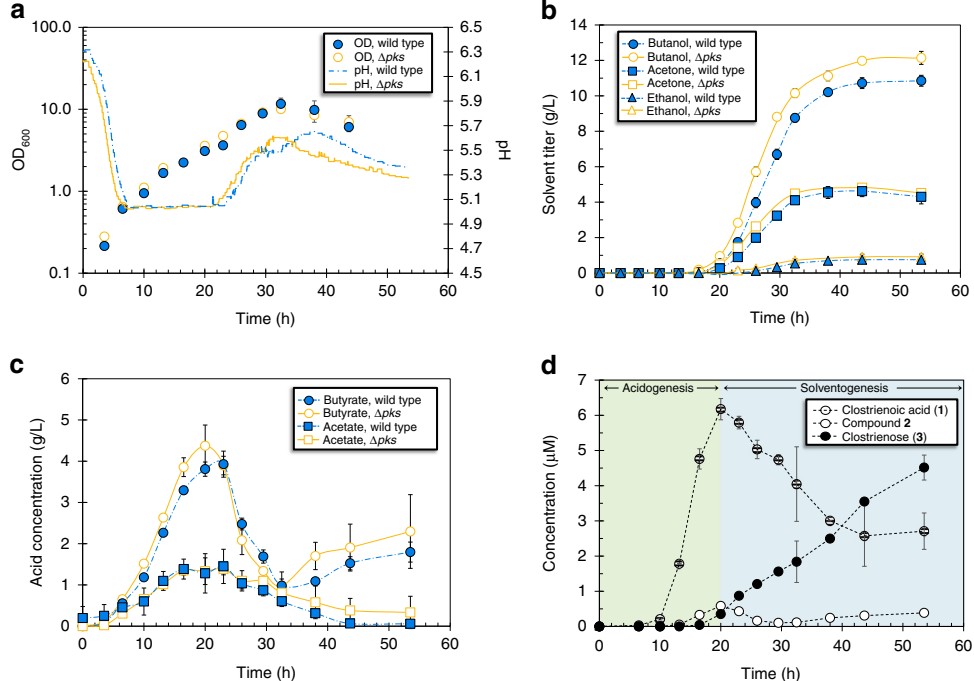

**Fig. 2** Time-course batch fermentation evaluation of wild-type *C. acetobutylicum* and Δ*pks*. **a** Culture density (measured as optical density at 600 nm, OD$_{600}$) (left axis) and pH (right axis) from batch fermentations of wild type and Δ*pks*. **b** Production of acetone, butanol, and ethanol from batch fermentations of wild type and Δ*pks*. **c** Production of acetate and butyrate from batch fermentations of wild type and Δ*pks*. **d** Production of compounds **1**–**3** from batch fermentation of wild-type *C. acetobutylicum* ATCC 824. As indicated, the green region represents the acid production phase of the fermentation (acidogenesis), while the blue region represents the solvent production phase (solventogenesis). Error bars represent the standard deviation of values from duplicate fermentations. Experiments were repeated at least three times independently

9). Similar derailment in the normal programmed steps to yield the same shunt products has been observed in other iterative type I PKSs such as LovB[28] and ApdA[29]. These results demonstrated that *ca_c3355* encodes a highly reducing type I PKS that functions iteratively to condense malonyl-CoA monomers. Furthermore, these results indicated that additional biosynthetic enzymes are needed for generating **1** and **3**, and their encoding genes are located elsewhere on the genome of *C. acetobutylicum*.

**Impact of *pks* gene on ABE fermentation.** To determine whether polyketide production influenced ABE fermentation, we compared the batch fermentation performance of wild-type *C. acetobutylicum* and Δ*pks* (Fig. 2a–c; Supplementary Fig. 10). While both strains displayed similar growth curves and the expected acidogenic and solventogenic phases, Δ*pks* showed stronger butanol production with ~10% increases in both butanol titer and productivity relative to wild type (Fig. 2a, b). Perhaps related to the difference in butanol production, the butyrate concentration profiles also differed for Δ*pks* and wild type; in the Δ*pks* culture, butyrate was produced more rapidly and reached a higher concentration during acidogenesis (0–20 h), and was re-assimilated earlier during the transition to solventogenesis (20–23 h) (Fig. 2c). This was also reflected in the pH profiles of the two strains, with the fall and rise of the culture pH (corresponding to the changes in metabolism) occurring earlier for Δ*pks* (Fig. 2a). Although less apparent, small increases in acetone and ethanol production were also observed for Δ*pks* relative to wild type (Fig. 2b). These results established a clear link between the *pks* gene and ABE fermentation, with improvements in solvent production associated with deletion of the *pks* gene.

To further probe the relationship between the production of the three polyketides and the ABE fermentation profile, we obtained production time-course profiles of **1**–**3** for wild-type

*C. acetobutylicum* (Fig. 2d). Maximum levels of **1** (clostrienoic acid) and **2** were observed during the same period as maximum butyrate/acetate concentrations (20–23 h), while the production of **3** (clostrienose) and butanol/acetone initiated at approximately the same time (~16 h) and increased for the remainder of the fermentation. These results suggest a direct, although currently unclear, link between polyketide production and the ABE fermentation phases, with the production of **1** and **2** associating with acidogenesis, and the production of **3** associating with solventogenesis. We propose that **1** and **2** are biosynthetic intermediates of the end product **3**, and feeding studies with **1** and **2** showed that these compounds were readily converted to **3** in cultures of *C. acetobutylicum* Δ*pks*.

**Polyketides affect sporulation and granulose accumulation.** To better understand the broader biological impacts of polyketide production on *C. acetobutylicum* metabolism and physiology, we performed a transcriptome comparison of Δ*pks* and wild-type strains using RNA-Seq. Analysis of samples taken at early stationary phase revealed a total of 392 genes that were differentially expressed (expression fold change > 2.0, *p*-value < 0.003) between Δ*pks* and wild type, with 282 genes downregulated and 110 genes upregulated in Δ*pks*. STRING network analysis[30] showed that the expression of genes related to four major cellular processes was downregulated in Δ*pks*, including sporulation (33 genes), carbohydrate transport (21 genes), carbohydrate metabolism (33 genes), and carbohydrate transfer (11 genes), and the expression of genes related to four other major cellular processes was upregulated in Δ*pks*, including sulfur metabolism (12 genes), cofactor biosynthesis (6 genes), stress response (8 genes), and amino-acid transport (8 genes) (Fig. 3; Supplementary Data 1). It is notable that no significant difference in the expression of the key solventogenic genes (including the crucial *sol* operon[31])

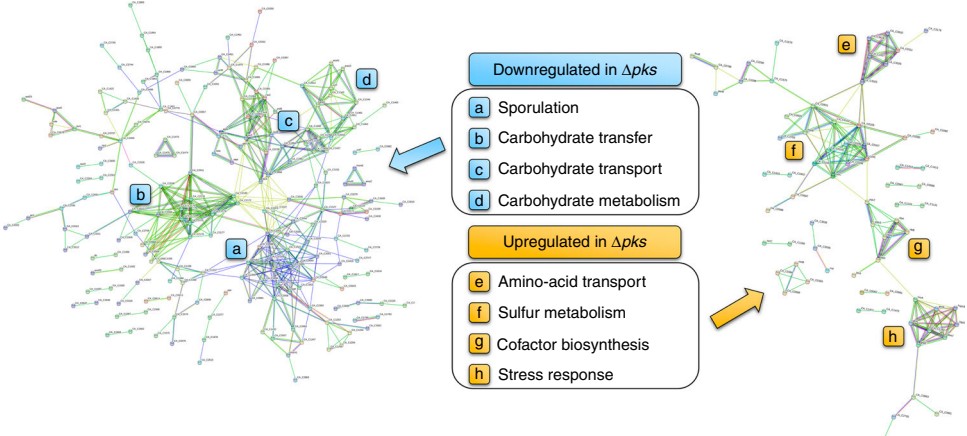

**Fig. 3** RNA-Seq comparison of wild-type *C. acetobutylicum* and Δ*pks*. STRING network analysis of genes predicted to be transcriptionally downregulated (left) and upregulated (right) by RNA-Seq in Δ*pks* fermentation culture relative to wild type. Nodes represent differentially expressed genes, and lines represent predicted connections between genes including shared functional pathways of encoded proteins, chromosomal proximity, co-occurrence of genes in other organisms, co-expression of genes, and protein homology. For clarity, only genes with at least one predicted connection are shown. Four major concentrations of connected nodes were observed for each group (indicated by **a**–**h**), signifying the cellular pathways most affected by deletion of the *pks* gene. Detailed results from the RNA-Seq analysis are presented in Supplementary Data 1

was observed, suggesting that the difference in butanol production between Δ*pks* and wild type was not due to the direct transcriptional regulation of solventogenic genes by the polyketides. Consistent with the higher solvent production observed for Δ*pks*, the major pathways upregulated in Δ*pks* (particularly class I, III, and IV heat shock response machinery) have previously been associated with improved solvent tolerance in *C. acetobutylicum*[14,32,33].

The sporulation genes downregulated in Δ*pks* include those encoding late-stage sporulation proteins such as spore coat and germination proteins, and the sporulation-specific sigma factor K (σ[K]), one of the core regulators of sporulation in *C. acetobutylicum*[34]. Notably, transcription of the gene encoding the well-known master regulator of sporulation and solvent production, Spo0A[35], was not significantly affected in Δ*pks*. To determine whether reduced sporulation was an observable property of Δ*pks* as suggested by the RNA-Seq analysis, sporulation assays in both liquid and solid media were performed for wild-type *C. acetobutylicum* and Δ*pks*. Indeed, a significant decrease in sporulation was observed for Δ*pks*, with spore formation decreasing by 3–4 orders of magnitude in liquid culture (Fig. 4a), and by ~2 orders of magnitude on solid media (Supplementary Fig. 11). Furthermore, the level of sporulation for Δ*pks* was partially restored when Δ*pks* culture broth was supplemented with clostrienose (**3**) (Fig. 4a). Since granulose biosynthesis and accumulation is related to the sporulation cycle, we then performed standard granulose accumulation assays using iodine staining. Colonies which have accumulated granulose are expected to stain dark brown/purple following exposure to iodine vapor, with darker staining indicating higher granulose production. Following the trend observed for sporulation, a significant decrease in granulose accumulation was observed for Δ*pks* (little to no staining), and the addition of **3** to Δ*pks* culture completely restored granulose accumulation (even appearing to exceed levels observed for wild type) (Fig. 4b). These assays revealed that the polyketides are important, although not essential, for triggering both sporulation and granulose accumulation in wild-type *C. acetobutylicum*.

Interestingly, *pks* inactivation also affected colony morphology of *C. acetobutylicum*. While wild-type colonies were relatively flat in elevation and featured a distinctive "spore center" in the middle of the colony, Δ*pks* colonies were distinctively raised in elevation and featured a highly textured surface with no distinguishable spore center (Fig. 4c). This behavior suggested that one or more of the polyketides might act as surfactants to permit colony spreading over the agar surface, as self-produced surfactants are known to permit surface motility in some microbes[36]. To test this, we performed standard assays for surfactant activity (oil spreading and drop collapse assays) using pure clostrienose (**3**), given that this compound displays some structural similarities to known non-ionic surfactants such as *n*-dodecyl-β-D-maltosides[37] and rhamnolipids[38]. As shown in Supplementary Tables 3 and 4, clostrienose displayed weak surfactant activity at relatively high concentrations (100 µM), but little to no surfactant activity near physiological concentrations (10 µM) according to the oil spreading assay. Surfactant activity was not observed using the drop collapse assay at either concentration of clostrienose (10 or 100 µM).

## Discussion

*Clostridium* is one of the largest bacterial genera, ranking second in size only to *Streptomyces*[39]. While members of *Streptomyces* are known to be prolific producers of secondary metabolites[40], only a handful of secondary metabolites have been discovered from *Clostridium*. However, recent genomic analysis has indicated that secondary metabolite gene clusters can be found among diverse members of this genus, prompting efforts to identify and characterize these "cryptic" secondary metabolites[6]. In this study, we identified a suite of polyketides (clostrienoic acid and clostrienose) from *C. acetobutylicum*, a well-studied solvent-producing anaerobe with no previously associated natural products.

Clostrienoic acid and clostrienose are biosynthesized by a predicted type I single-module PKS. Through in vitro reconstitution of the activity of purified PKS, we demonstrated that this megasynthase functions as a highly reducing iterative type I PKS. We propose that the PKS functions iteratively to generate a heptaketide intermediate, which is then modified by tailoring enzymes encoded elsewhere on the genome to yield clostrienoic acid. Two subsequent glycosylation events are proposed to install first the rhamnopyranoside group, followed by the galactofuranoside group (generating clostrienose) (Supplementary Fig. 12). The proposed sequential biosynthesis of the three polyketides is consistent with the production timing of **1**–**3** observed in batch fermentation of *C. acetobutylicum*. Although galactofuranose has

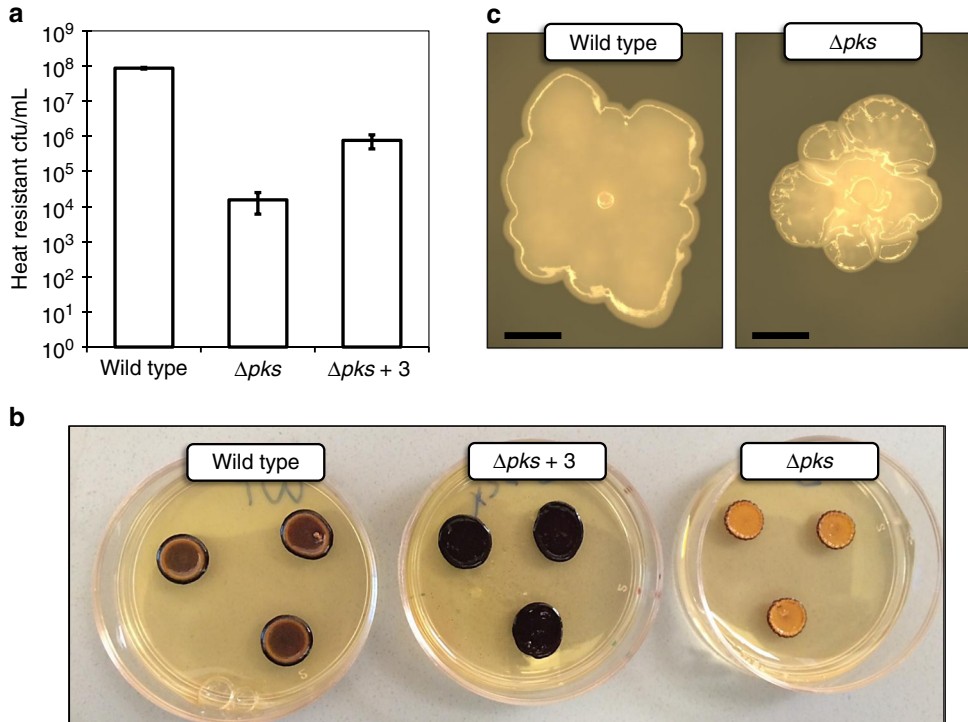

**Fig. 4** Phenotype comparison of wild-type *C. acetobutylicum* and Δ*pks*. **a** Sporulation assay results for liquid cultures of wild type, Δ*pks*, and Δ*pks* supplemented with clostrienose (**3**). Values represent the number of heat-resistant colony forming units per mL following incubation in liquid CBM-S medium for 5 days. Error bars represent the standard deviation of biological triplicate cultures. Experiments were repeated at least three times independently. **b** Granulose accumulation assay results for wild type, Δ*pks*, and Δ*pks* supplemented with clostrienose (**3**). Patches containing high levels of granulose are expected to stain dark brown/ purple. **c** Dissecting microscope images of individual wild-type and Δ*pks* colonies. Black scale bar shows 1 mm in length

previously been detected in cultures of *C. acetobutylicum*[41] and the predicted biosynthetic genes for both sugar moieties are present on the *C. acetobutylicum* genome, the dedicated glycosyltransferases employed for the production of **3** are unclear. Due to the apparent non-clustered nature of genes encoding the enzymatic partners of PKS and the observation that hundreds of genes were upregulated along with *ca_c3355* at early stationary phase in *C. acetobutylicum*, extensive future work is needed to identify additional biosynthetic genes and elucidate the complete biosynthetic pathway of **3**.

Considering all of our results, comparison of wild-type *C. acetobutylicum* and Δ*pks* along with chemical complementation of Δ*pks* strongly suggest that the identified polyketide, clostrienose (**3**), serves as a signaling molecule that regulates granulose accumulation and sporulation in *C. acetobutylicum*. For clarity, we define "signaling molecule" here as a biologically derived secreted small molecule, which acts to directly or indirectly regulate gene expression (beyond that related to metabolizing or detoxifying the molecule)[42,43]. Although cellular communication by signaling small molecules has been reported in a variety of Gram-negative and Gram-positive bacteria, with well-known examples such as *N*-acyl-homoserine lactones[44], 4-quinolones[45], γ-butyrolactones[46], and autoinducer (AI)-2[47], few signaling small molecules have been reported from *Clostridium*. In addition to clostrubin, which helps the pathogen *C. puniceum* access aerobic territory[9], a putative AI-2-like system was reported to be involved in stimulating toxin production in *C. perfringens*[48]. Furthermore, gene clusters with homology to the well-studied accessory gene regulator (*agr*) quorum sensing system have been shown to be important for cellular processes such as toxin production[49–52], biofilm formation[49], and sporulation[52,53] in several *Clostridium* species, although none of these studies identified the molecular structure of the presumed

bioactive small molecules. This work provides solid evidence that cellular differentiation can be regulated by secondary metabolites in *Clostridium*, and has revealed the molecular identity of the signal responsible for this behavior.

Although clostrienose displayed only weak surfactant activity near physiological concentrations observed under batch fermentation conditions (<6 μM), the possible role of clostrienose as both a biosurfactant as well as a signaling molecule draws interesting parallels to previously studied biosurfactants, which are known to act as signaling molecules to regulate gene expression. In the well-studied case of surfactin (widely classified as a quorum sensing signaling molecule[54–56]), this self-generated biosurfactant is known to regulate gene expression necessary for triggering extracellular matrix (ECM) production and efficient sporulation in *Bacillus subtilis*[54,57]. Interestingly, surfactin is able to alter gene expression by forming membrane pores, which selectively leak potassium ions. The decrease in intracellular potassium is sensed by the membrane histidine kinase KinC, which then activates the regulatory circuit leading to ECM production[54,55]. In another case, *N*-acyl-homoserine lactones (AHLs, a well-known class of signaling molecules[58,59]) natively produced by the bacterium *Rhizobium etli* were shown to act as both biosurfactants and quorum sensing molecules to promote surface colonization[60]. These examples illustrate that self-produced biosurfactants are not restricted to their physiochemical role in decreasing surface tension, and can also serve to regulate gene expression by acting as signaling molecules for some microbes. For the case of *C. acetobutylicum*, it would be interesting to determine if clostrienose features a mode of action similar to that of surfactin.

Although extensive future work is needed to characterize the full regulatory network through which clostrienose controls cellular differentiation, results from our RNA-Seq analysis suggest

that transcriptional regulation of the sporulation-specific sigma factor, $\sigma^K$, may play an important role in the mode of action. Based on our RNA-Seq results, we observed the downregulation of 33 genes related to sporulation, in particular, genes related to late endospore development (stages IV and V of the sporulation cycle[12]) and spore germination in $\Delta pks$ relative to wild type (Supplementary Data 1). Of the well-characterized regulators of sporulation in *C. acetobutylicum* ATCC 824 (including Spo0A, $\sigma^H$, $\sigma^E$, $\sigma^F$, $\sigma^G$, and $\sigma^K$)[12,61–63], only the gene encoding $\sigma^K$ (*sigK* [*ca_c1689*]) was significantly downregulated in $\Delta pks$ at the time of analysis (26 h post inoculation). A putative sigma factor encoded by *ca_p0157* (believed to be involved in regulating very late-stage sporulation based on a prior study[64]) was also significantly downregulated in $\Delta pks$, suggesting that transcription of this regulator may be under the control of $\sigma^K$. It was reported recently that $\sigma^K$ performs two developmentally separated roles in *C. acetobutylicum* ATCC 824, one in early sporulation and one in late sporulation[34]. In early sporulation, $\sigma^K$ is important for upregulation of Spo0A, the master regulator of sporulation and solvent production. For this early role, *sigK* likely requires transcriptional activation by $\sigma^E$. Since Spo0A is important for initiating both solventogenesis and sporulation, *sigK* deletion results in low solvent, non-sporulating cultures. In late sporulation, $\sigma^K$ is important for stage IV spore development, including assembly of the spore coat. For this late role, $\sigma^K$ activation is $\sigma^G$ dependent, likely through contributions of the $\sigma^G$ dependent genes *spoIVFB* [*ca_c1253*] and *spoIVB* [*ca_c2072*], which are proposed to be required for post-translational processing of pro-$\sigma^K$ to the mature $\sigma^K$ form. Considering our results, we hypothesize that transcriptional downregulation of *sigK* in $\Delta pks$ at this stage of the fermentation is relevant to the role of $\sigma^K$ in late-stage spore development, as $\Delta pks$ displayed severe reductions in observed sporulation rates, but ABE production was not decreased as would be expected for weakened early-stage $\sigma^K$ activity. As we expect that some or all of sporulation genes downregulated in $\Delta pks$ are members of the $\sigma^K$ regulon, this suggests that clostrienose may have some role in stimulating late-stage $\sigma^K$ activity, and thus, late-stage sporulation. Given what is known about the regulatory relationships between well-characterized sporulation sigma factors in *C. acetobutylicum*[12], we hypothesize that clostrienose may act to regulate late-stage $\sigma^K$ activity through an unknown pathway between $\sigma^G$ and $\sigma^K$, possibly involving contributions in post-translational regulation of $\sigma^K$ from the $\sigma^G$-dependent enzymes SpoIVFB and SpoIVB (the encoding genes of which were also downregulated in $\Delta pks$) (Supplementary Data 1). While extensive follow up work is required to validate this hypothesis, this proposed mechanism could help to explain the means by which $\sigma^G$ activates late $\sigma^K$ activity in *C. acetobutylicum*, which is currently unknown[12].

Solventogenic strains of *Clostridium* (such as *C. acetobutylicum*) were employed for industrial solvent production as early as 1916, and were eventually the source of 66% of US butanol production in 1945[10]. Although industrial operations of this process largely ceased during the 1950s due to the growth of the petrochemical industry, ABE fermentation has recently gained renewed interest given the wide range of agricultural feedstocks, which can be converted to various commodity chemicals and potential biofuels using this process[65]. However, some drawbacks still exist that prevent the widespread use of ABE fermentation, such as low solvent titer/productivity due to solvent toxicity, and unfavorable cellular differentiation[66]. Rather than pursuing a traditional metabolic engineering strategy that focuses on the core metabolic pathway for solvent production, our work showcases an alternative approach by manipulating the secondary metabolism of the organism to improve traits significant for industrial ABE fermentation performance. In particular, given

that both granulose biosynthesis and sporulation are undesirable traits for industrial fermentation (as granulose accumulation results in reduced solvent yields, and metabolically inactive spores do not contribute to solvent production)[66], the reduced granulose accumulation and sporulation associated with $\Delta pks$ represent improved industrial traits. Furthermore, although none of the solvent producing genes were upregulated in $\Delta pks$ relative to wild type, both butanol titer and productivity were increased in $\Delta pks$. This may be explained by the decreased commitment of cells to sporulation in $\Delta pks$ (yielding a higher proportion of cells capable of solvent production), as well as the upregulation of cellular machinery related to butanol stress and adaptation as indicated by transcriptomic analysis.

In summary, we have discovered a family of novel polyketides that are biosynthesized by a highly reducing iterative type I PKS in *C. acetobutylicum* ATCC 824. In addition to the type II PKS-derived clostrubin, our work provides the second example of polyketide metabolites from a strictly anaerobic bacterium, and encourages continued efforts in exploring the uncharted terrain of secondary metabolites in the anaerobic world. We have further shown that the newly identified polyketides are important for stimulating sporulation and granulose accumulation in *C. acetobutylicum*, adding to the extremely limited inventory of known signaling molecules used by *Clostridium* to control cellular physiology and metabolism. Furthermore, this work has yielded an engineered strain of *C. acetobutylicum* with improved traits for industrial ABE fermentation (reduced sporulation, reduced granulose accumulation, and increased butanol titer and productivity), demonstrating a novel strategy of manipulating secondary metabolism as a means of improving this important renewable bioprocess.

## Methods

**Bacterial strains and media.** *C. acetobutylicum* ATCC 824 was cultured in an anaerobic chamber (Coy Laboratory Products) containing an atmosphere of 97% nitrogen and 3% hydrogen. 2xYTG medium[67] contained 16 g/L tryptone, 10 g/L yeast extract, 5 g/L NaCl, and 10 g/L glucose (unless noted otherwise) with the pH adjusted to 5.2 for liquid media, and 5.8 for solid media (also containing 15 g/L agar). Clostridial growth medium (CGM)[68] contained 30 g/L glucose (unless noted otherwise), 6.25 g/L yeast extract, 2.5 g/L ammonium sulfate, 1.25 g/L NaCl, 2.5 g/L asparagine, 0.95 g/L monobasic potassium phosphate, 0.95 g/L dibasic potassium phosphate, 0.5 g/L magnesium sulfate heptahydrate, 13 mg/L manganese sulfate heptahydrate, and 13 mg/L iron sulfate heptahydrate with the pH adjusted to 6.4. P2 medium[69] contained 80 g/L glucose (unless noted otherwise), 1 g/L yeast extract, 2.2 g/L ammonium acetate, 0.5 g/L potassium phosphate monobasic, 0.5 g/L potassium sulfate dibasic, 0.2 g/L magnesium sulfate heptahydrate, 1 mg/L para-aminobenzoic acid, 1 mg/L thiamine hydrochloride, 10 μg/L biotin, 10 mg/L manganese sulfate heptahydrate, 10 mg/L ferrous sulfate heptahydrate, and 10 mg/L NaCl with the pH adjusted to 6.4. Clostridial basal medium (CBM)[70] contained 10 g/L glucose, 0.5 g/L monobasic potassium phosphate, 0.5 g/L dibasic potassium phosphate, 4 g/L tryptone, 0.2 g/L magnesium sulfate heptahydrate, 10 mg/L manganese sulfate heptahydrate, 10 mg/L ferrous sulfate heptahydrate, 1 mg/L para-aminobenzoic acid, 1 mg/L thiamine hydrochloride, and 2 μg/L biotin with the pH adjusted to 6.9. For solid CGM plates, 15 g/L agar was added. CBM-S (used for liquid sporulation assays) was identical to CBM except 50 g/L glucose was used, and 5 g/L CaCO$_3$ was added just prior to inoculation of cultures. *E. coli* TOP10 (Thermo Fischer Scientific) was grown in Luria-Bertani (LB) medium at 37 °C. For the appropriate *Clostridium* strains, culture media was supplemented with erythromycin (Ery; 40 μg/mL for solid media, 80 μg/mL for liquid media) and/or thiamphenicol (Th; 5 μg/mL for solid and liquid media). Kanamycin (Kan; 60 μg/mL) or chloramphenicol (Cm; 25 μg/mL) were added to *E. coli* culture media as indicated. *Clostridium* and *E. coli* strains were maintained as 20% v/v glycerol stocks stored at −80 °C.

**Plasmid construction.** Oligonucleotides were provided by Integrated DNA Technologies (Supplementary Table 5). Phusion polymerase (NEB) was used for all PCR reactions. For isolation of genomic DNA from *C. acetobutylicum* ATCC 824, an alkaline lysis method was used[71]. *C. acetobutylicum* was cultured overnight in 2xYTG medium to stationary phase (OD$_{600}$ > 2.0), at which point 10 mL of culture was centrifuged at 3500×*g* for 15 min (room temperature). The supernatant was discarded and the cell pellet was resuspended in 5 mL SET buffer (75 mM NaCl, 25 mM EDTA pH 8.0, 20 mM Tris-HCl, pH 7.5). Lysozyme was added to a final concentration of 2 mg/mL, and the solution was gently mixed. The mixture was

incubated at 37 °C for 60 min with gentle mixing performed every 15 min. Following the incubation period, 660 μL of lysis buffer (1 M NaOH, 10% w/v SDS) was added, and the solution was thoroughly mixed. Proteinase K was added to a final concentration of 0.5 mg/mL, and the solution was incubated at 55 °C for 1 h. Following this incubation period, an equal volume of phenol:chloroform (1:1) was added, and the solution was mixed by inversion for 5 min. The solution was then centrifuged at 3500×g for 15 min (room temperature), and the upper aqueous phase was discarded by pipetting. To the organic phase, 3 M sodium acetate was added (10% v/v in final solution). To precipitate genomic DNA, 2 volumes of ethanol were added and the solution was mixed. Precipitated genomic DNA was collected using a glass hook, and washed with 70% ethanol. The washed DNA was then dried over nitrogen gas for 15 min, resuspended in low TE buffer (10 mM Tris, 0.1 mM EDTA, pH 8.0), and dissolved by incubation at 50 °C.

For constructing plasmid pKO_mazF_mod (which would later serve as a template for pKO_pks), primers pKON_Fo and pKON_Ro were used to PCR amplify a 5.0 kb region from the pKO_mazF template[15]. This step was necessary to remove a 677 bp region from the pKO_mazF backbone, which we were unable to amplify via PCR. The 5.0 kb PCR product was gel purified, digested at the 5′ and 3′ ends using PshAI, ligated to form pKO_mazF_mod, and transformed into E. coli TOP10. Transformant clones were screened by purified plasmid test digestion, and Sanger sequencing was used to confirm the sequence of the final pKO_mazF_mod clone. For constructing plasmid pKO_pks (for deletion of the pks gene from C. acetobutylicum), a 3.8 kb region containing colE1, repL, bgaR, and mazF were PCR amplified from pKO_mazF_mod using primers pKO_F and pKO_R. Additionally, primers UHR_Fo and UHR_Ro, and DHR_Fo and DHR_Ro were used to PCR amplify 1 kb regions representing the upstream and downstream homologous regions (UHR and DHR) flanking the pks gene (CA_C3355) from a C. acetobutylicum ATCC 824 genomic DNA template. The chloramphenicol/ thiamphenicol resistance gene and constitutive promoter ($P_{ptb}$ from C. acetobutylicum) were PCR amplified from pKO_mazF_mod using primers CMR_F and CMR_R (1.1 kb region). These four PCR products were ligated via Gibson assembly and transformed into E. coli TOP10. Transformant clones were screened by purified plasmid test digestion, and Sanger sequencing was used to confirm the sequence of the final pKO_pks clone. For constructing plasmid pAN3[15] (necessary for methylation of pKO_pks and pCKO_pks prior to transformation into C. acetobutylicum), a 6.4 kb region from plasmid pAN1[72] was amplified using primers pAN1_F and pAN1_R, and a 1.0 kb region containing the kanamycin resistance gene from vector pCOLA_Duet was amplified using primers KAN_Fo and KAN_Ro. The gel extracted PCR products were ligated via Gibson assembly and transformed in to E. coli TOP10. Transformant clones were screened by purified plasmid test digestion, and Sanger sequencing was used to confirm the sequence of the final pAN3 clone. For constructing plasmid pCKO_pks (for expression of the pks gene under the constitutive crotonase [crt] promoter from C. acetobutylicum), primers CPKS_Fo and CPKS_Ro were used to PCR amplify the 5.4 kb C. acetobutylicum ATCC 824 pks gene (CA_C3355) with appropriate overhangs for Gibson assembly. The pWIS_empty vector[71] backbone (containing the constitutive crt promoter upstream of the multiple cloning site) was PCR amplified using primers pWIS_F and pWIS_R yielding a 5.0 kb product. The two gel purified PCR products were ligated via Gibson assembly and transformed into E. coli TOP10. Transformant clones were screened by purified plasmid test digestion, and Sanger sequencing was used to confirm the sequence of the final pCKO_pks clone.

For constructing plasmid pET24b_pks, the 5.4 kb C. acetobutylicum ATCC 824 pks gene (CA_C3355) was amplified from C. acetobutylicum genomic DNA using primers PKS_Fo and PKS_Ro. Doubly digested vector pET24b (EcoRI/XhoI digested) was ligated with doubly digested pks PCR product (EcoRI/XhoI digested), and the ligation product was transformed into E. coli TOP10. Transformant clones were screened by purified plasmid test digestion, and Sanger sequencing was used to confirm the sequence of the final pET24b_pks clone.

**Electro-transformation of C. acetobutylicum.** Prior to transformation into C. acetobutylicum, vector pKO_pks was co-transformed with pAN3 into E. coli TOP10 via electroporation. This procedure permitted methylation of pKO_pks necessary for overcoming the native restriction-modification system active in C. acetobutylicum[72]. Plasmid purification of E. coli pKO_pks/pAN3 liquid culture was performed, and the resulting plasmid mixture was used for electroporations of C. acetobutylicum using the previously published method[72] which we detail here. To prepare electrocompetent cells, a single colony of C. acetobutylicum from a 2xYTG plate (more than 1 week old) was heat shocked at 80 °C for 10 min, and used to inoculate 10 mL CGM. After incubating overnight at 37 °C and reaching mid-exponential phase ($OD_{600}$ 0.4–0.9), 6 mL culture was used to inoculate 54 mL fresh 2xYTG. This subculture was incubated at 37 °C until reaching $OD_{600}$ 1.2 (~5 h), at which point the subculture was centrifuged at 3500×g for 20 min (4 °C). Following removal of the supernatant, the cell pellet was resuspended in 20 mL ice-cold electroporation buffer (EPB; 270 mM sucrose, 5 mM $NaH_2PO_4$, pH 7.4). The resuspended cells were centrifuged at 3500×g for 10 min (4 °C), the supernatant was discarded, and the cell pellet was resuspended in 20 mL ice-cold EPB. After a final pelleting by centrifugation at 3500×g for 10 min (4 °C), the supernatant was discarded, and the pellet was resuspended in 2.3 mL ice-cold EPB. This final resuspension was used as the stock of electrocompetent cells. Electro-transformations were performed by first mixing (over ice) 500 μL competent cells

and ~20 μL of plasmid DNA solution (1–5 μg total) to be transformed. The mixture was transferred to a 0.4 cm electroporation cuvette, and allowed to incubate on ice for 15 min. Electroporations were performed with the following parameters: voltage, 2000 V; capacitance, 25 μF; resistance, infinite Ω. Following electroporation, samples were resuspended in 1 mL 2xYTG and transferred to a tube containing 9 mL 2xYTG. After mixing by inversion, the cultures were allowed to recover at 37 °C for 4 h. The recovery cultures were centrifuged at 3500×g for 15 min (room temperature), and the supernatant was discarded. The cell pellet was resuspended in 500 μL 2xYTG, and 100 μL was plated onto solid 2xYTG plates supplemented with the appropriate antibiotic. The plate was incubated at 37 °C for 2–3 days, and transformant colonies were subjected to PCR-based verification.

**Deletion of pks gene and genetic complementation.** Targeted KO of the pks gene (ca_c3355) in C. acetobutylicum ATCC 824 was achieved using the previously published method[15]. In detail, 5 μg of methylated pKO_pks/pAN3 plasmid mixture was transformed into C. acetobutylicum using the method described above. Following recovery in liquid 2xYTG medium for 4 h, the cell pellets were collected by centrifugation (3500×g, 15 min, room temperature), resuspended in 0.5 mL of fresh liquid 2xYTG, and 100 μL of the resuspended cell culture was plated on solid 2xYTG + 5 μg/mL Th + 40 mM β-lactose plates. Under these plating conditions, only cells which have undergone the desired double crossover homologous recombination event are expected to survive. Counterselection of the vector backbone is provided by the lactose-inducible promoter ($P_{bgaL}$), which drives the toxin gene mazF on the pKO_pks vector backbone. Following this plating procedure, roughly 10 colonies were observed on the 2xYTG + 5 μg/mL Th + 40 mM β-lactose plates. Of these 10 colonies, four were twice restreaked and subjected to colony PCR verification. Four sets of primers were used as the basis of colony PCR verification, as detailed in Supplementary Fig. 2.

For generating the pks genetic complementation strain [Δpks (pCKO_pks)], electroporation of Δpks C. acetobutylicum with a pCKO_pks/pAN3 plasmid mixture was performed as described above. Following electroporation and recovery, cultures were plated on solid 2xYTG + 5 μg/mL Th + 40 μg/mL Ery media. After twice restreaking on solid 2xYTG + 5 μg/mL Th + 40 μg/mL Ery media, potential colonies harboring pCKO_pks were screened via colony PCR using primers pCKO_F and pCKO_R.

**LC-HRMS metabolomic analysis.** For untargeted metabolomic comparisons of wild-type and Δpks C. acetobutylicum, single colonies of each strain were heat shocked at 80 °C for 10 min and used to inoculate 10 mL of liquid CGM (30 g/L glucose). Following overnight incubation (stagnant) until reaching $OD_{600}$ ~1.0, these cultures were used to inoculate 10 mL of liquid CGM (80 g/L glucose) with a 10% inoculum for subculturing. After ~5 h ($OD_{600}$ ~1.0) of stagnant growth, the subcultures were used to inoculate quadruplicate flask fermentations (70 mL CGM, 80 g/L glucose + 5 μL Antifoam 204, 3% inoculum) agitated via magnetic stir bars. Calcium carbonate (6 g/L) was supplemented to the fermentation media for pH buffering. All culturing was performed at 37 °C. About 5 μg/mL Th was included in all cultures of Δpks with the exception of the final fermentation culture, as certain antibiotics are known to perturb the ABE fermentation phases. Samples of fermentation broth (1 mL) from each replicate were taken during early stationary phase and extracted with 3 mL of 2:1 chloroform methanol. The mixtures were vortexed, separated via centrifugation (2700×g, 10 min), and the bottom chloroform-rich layer was transferred to a glass vial. These organic extracts were dried with nitrogen gas, resuspended in 100 μL methanol, and 10 μL was injected onto an Agilent Technologies 6520 Accurate-Mass QTOF LC-MS instrument fitted with an Agilent Eclipse Plus C18 column (4.6 × 100 mm). A linear gradient of 2–98% $CH_3CN$ (vol/vol) over 40 min in $H_2O$ with 0.1% formic acid (vol/vol) at a flow rate of 0.5 mL/min was used. The metabolomic analysis platform XCMS[16] (The Scripps Research Institute) was used to compare the metabolomes of wild-type and Δpks strains based on the quadruplicate LC-HRMS data. MS peaks unique to Δpks (Fig. 1a) were identified using the following parameters: p-value < 0.01, fold change > 10.0, peak intensity > 5000.

**Bioreactor fermentations.** To compare ABE fermentation profiles of wild-type and Δpks C. acetobutylicum, bioreactor fermentations were carried out in DASGIP Bioreactors (4 × GPI 100 Vessels, DASGIP Bioblock System) with 500 mL working volumes. Overnight cultures (10 mL CGM, 30 g/L glucose, stagnant, 34 °C) inoculated with heat shocked individual colonies of C. acetobutylicum were cultured until reaching $OD_{600}$ ~1. A 10% inoculum was then used to start a subculture (30 mL P2 media, 80 g/L glucose, stagnant, 34 °C), and the subculture was incubated until reaching $OD_{600}$ ~1. The 30 mL subcultures were then aseptically transferred into individual DASGIP Bioreactors pre-loaded with 500 mL P2 medium (80 g/L glucose, 100 μL Antifoam 204, 34 °C). The fermentations were allowed to proceed for 54 h with periodic sampling for optical density measurements, fermentation product analysis, and quantification of compounds **1**, **2**, and **3**. The temperature was maintained at 34 °C throughout the fermentation, agitation was provided by stirring at 200 rpm, and the pH was maintained above 5.0 via automatic addition of 3 M $NH_4OH$. To maintain anaerobic conditions, oxygen-free nitrogen gas was sparged at a rate of 2 sL/h for the duration of the fermentation. For quantification of compounds **1**, **2**, and **3**, 1 mL of fermentation broth was

mixed with 3 mL ethyl acetate, vortexed, separated via centrifugation (2700×*g*, 10 min, room temperature), and the upper organic layer isolated. The organic layer was dried by rotary evaporation, resuspended in 200 μL methanol, and 20 μL was injected onto an Agilent Technologies 6120 Quadrupole LC-MS (with DAD) instrument fitted with an Agilent Eclipse Plus C18 column (4.6 × 100 mm). A linear gradient of 2–98% CH$_3$CN (vol/vol) over 40 min with H$_2$O with 0.1% formic acid (vol/vol) at a flow rate of 0.5 mL/min was used. Compounds **1**, **2**, and **3** were identified by UV absorption (240 nm) as demonstrated in Fig. 1b, and were quantified by the integrated peak area (absorbance at 240 nm).

**Fermentation analytical procedures.** A spectrophotometer was used to determine cell densities by measuring the optical density at 600 nm (OD$_{600}$). A Shimadzu Prominence UFLC system fitted with a Biorad Aminex HPX-87H column (300 mm × 7.8 mm) was used to analyze *C. acetobutylicum* fermentation broth for the concentration of glucose and fermentation products (acetate, butyrate, lactate, acetone, butanol, and ethanol). Samples of fermentation broth (1 mL) were first pelleted by centrifugation at 10,000×*g* for 3 min, followed by filtration of the supernatant using a 0.22 micron PVDF syringe filter. Samples of filtered supernatant (20 μL) were injected onto the UFLC system with 0.01 N sulfuric acid mobile phase flowing at 0.7 mL/min, column temperature of 35 °C, and were chromatographed for 35 min. Analytes were detected by use of a refractive index detector (used to quantify concentrations of glucose, acetate, lactate, butanol, and ethanol) and a diode array detector (used to quantify concentrations of butyrate (208 nm) and acetone (265 nm)).

**RNA isolation and RNA-Seq analysis.** Samples (10 mL) of fermentation broth were taken in biological triplicate from bioreactor fermentations of wild-type and Δ*pks C. acetobutylicum* 26 h post inoculation. The samples were centrifuged (4000×*g*, 10 min, 4 °C) and the pellets were resuspended and stored in RNAprotect Cell Reagent (Qiagen). Total RNA was extracted using an RNeasy Mini Kit (Qiagen) according the manufacturer's instructions. An on-column DNase treatment was performed using DNase I (RNase-free) (NEB). RNA quality control, library construction, and library sequencing were performed by the University of California-Berkeley QB3 Functional Genomics Laboratory and Vincent J. Coates Genomic Sequencing Laboratory. RNA quality and concentration was assessed using a nanochip on an Agilent 2100 Bioanalyzer. Bacterial 16S and 23S rRNA was removed using a RiboZero Kit (Illumina). The resulting messenger RNA (mRNA) was converted to an RNA-Seq library using an mRNA-Seq library construction kit (Illumina). RNA library sequencing was performed on an Illumina HiSeq4000 with 50 bp single end reads. Sequencing reads (50 bp) were processed and mapped to the *C. acetobutylicum* ATCC 824 genome (NCBI accession NC_003030.1 [chromosome] and NC_001988.2 [megaplasmid]) using CLC Genomics Workbench 9.0 with default parameters. Reads that did not uniquely align to the genome were discarded. Differences in gene expression between wild-type and Δ*pks C. acetobutylicum* were calculated using the same software. Genes were considered differentially expressed with *p*-value < 0.003 (based on an unpaired *t*-test, *n* = 3) and |normalized fold change |> 2.0. False discovery rate corrections to *p*-values were calculated in CLC Genomics Workbench 9.0 using a published method[73]. Results of the RNA-Seq analysis are presented in Supplementary Data 1. STRING analysis[30] was performed to determine putative protein–protein interactions between the differentially expressed genes revealed by RNA-Seq analysis.

**Phenotype comparison assays.** Liquid sporulation assays were performed with minor modifications[74]. Samples were taken from biological triplicate liquid cultures after 5 days of incubation (30 mL CBM-S, 37 °C). The 20 μL samples were heat shocked (80 °C, 10 min), dilutions (10$^1$–10$^6$) were spotted on 2xYTG plates, and colonies were enumerated after 30 h of incubation (37 °C) to calculate the number of heat-resistant colony forming units (cfu/mL). For chemical complementation of Δ*pks* in the liquid sporulation assay, purified clostrienose (final concentration 3.5 μM) was supplemented in CBM-S cultures of Δ*pks C. acetobutylicum* at the time of inoculation. Since compound **3** was added as a concentrated solution in methanol, the equivalent volume of methanol (60 μL) was added to all other liquid sporulation assay cultures (wild-type and non-complemented Δ*pks*) to control for this effect.

For solid sporulation assays, a previously described method[75] was employed with some modifications. In detail, heat shocked (80 °C, 10 min) individual colonies were cultured in liquid media (10 mL CGM, 37 °C) for 24 h. Cultures were diluted by a factor of 10$^6$ and plated on solid CBM. Sampling was conducted 1, 2, 3, 4, and 6 days following initial plating. For sampling, three individual colonies were combined and thoroughly resuspended in 60 μL of liquid CGM. 20 μL of the resuspended colony mixture was then heat shocked (80 °C, 10 min), dilutions (10$^1$–10$^6$) were spotted on 2xYTG plates, and colonies were enumerated after 30 h of incubation (37 °C) to calculate the number of heat resistant colony forming units (cfu/colony). All samples were performed as biological triplicates.

Granulose accumulation assays were performed via iodine staining[53]. Mid-log phase (OD$_{600}$ ~0.6) liquid cultures (P2, 37 °C) were plated on solid 2xYTG medium with elevated glucose levels (50 g/L) to enable granulose production. The plates were incubated at 30 °C for 4 days, at which point they were stained by exposure to a bed of iodine crystals for 10 min. The plates were then allowed to destain for

10 min prior to imaging. For chemical complementation of Δ*pks* in the granulose accumulation assay, purified clostrienose (final plate concentration 4.0 μM) was embedded in solid 2xYTG plates prior to plating of Δ*pks* culture. Since clostrienose was added to 2xYTG plates as a concentrated solution in methanol (mixed into the molten media prior to solidification), the equivalent volume of methanol (6 μL) was added to all other 2xYTG plates used in the granulose accumulation assays to control for this effect.

Individual colonies of *C. acetobutylicum* cultured on CBM plates for 48 h (37 °C) were viewed and imaged using a Leica MZ16 F dissecting microscope fitted with a Leica DFC300 FX camera.

Oil spreading assays were performed as previously described[76–78]. In detail, A 400 mL glass beaker (7.5 cm inner diameter) was filled with 300 mL water, and a 10 μL droplet of paraffin lamp oil was added and allowed to spread into a thin film on the water surface over a 5 min period (~25 mm in diameter). Solutions of surfactin, purified clostrienose, and a potassium phosphate buffer control were prepared at the indicated pH and concentration (prepared in 10 mM potassium phosphate buffer). About 10 μL of sample was carefully added to the center of the oil film, and the effect on the oil film was observed. Samples with surfactant activity are expected to form stable circular clearing zones in the oil film, with more potent surfactant activity corresponding to larger clearing zones (or complete dispersion of the oil film for very-high-surfactant activity). Drop collapse assays were performed[77,79,80]. Circular microwells (8 mm inner diameter) within the polystyrene lid of a 96-microwell plate (12.7 × 8.5 cm) were coated with a thin layer of paraffin lamp oil by adding 2.0 μL of paraffin lamp oil to each well, and allowing the droplet to spread evenly over the microwell over a 1 h period. Samples of surfactin, purified clostrienose, and potassium phosphate buffer control were prepared as for the oil spreading assays. About 5 μL of sample was carefully added to the center of the microwell. After 5 min, the shape and degree of spreading of the droplet was observed. While buffer controls are expected to form stable beads on the microwell surface, samples containing surfactant are expected to collapse and spread over the microwell surface, with more potent surfactant activity corresponding to a higher degree of droplet spreading.

**Purification and in vitro analysis of PKS protein.** To purify the His$_6$-tagged PKS protein for in vitro analysis, pET24b_pks was transformed into *E. coli* BAP1. A single colony was inoculated into 10 mL LB + 50 μg/mL kanamycin (Kan) for overnight growth at 37 °C. About 7 mL of overnight culture was used to inoculate 700 mL LB + 50 μg/mL Kan, and the culture was shaken at 240 rpm and 37 °C until reaching OD$_{600}$ of 0.5. After icing the culture for 10 min, isopropyl thio-β-D-galactoside (IPTG) was added to a final concentration of 0.1 mM to induce protein expression, and the culture was incubated at 16 °C for 16 h. The cells were harvested by centrifugation (5500×*g*, 4 °C, 20 min), resuspended in 25 mL lysis buffer (25 mM HEPES, pH 8.0, 0.5 M NaCl, 5 mM imidazole), and lysed by homogenization over ice. Cell debris was removed by centrifugation (17,700×*g*, 4 °C, 60 min), and Ni-NTA agarose resin was added to the supernatant (2 mL/L culture). The mixture was nutated at 4 °C for 1 h, loaded onto a gravity flow column, and the PKS protein was eluted with increasing concentrations of imidazole in Buffer A (20 mM HEPES, pH 8.0, 1 mM DTT). Purified PKS protein was concentrated and buffer exchanged into Buffer A + 10% glycerol using a Vivaspin Centrifugal Concentrator. Aliquots of purified PKS protein were aliquoted and flash frozen in liquid nitrogen. The approximate protein yield was 5 mg/L (203 kDa).

A PKS loading assay with $^{14}$C-labeled substrate contained, in a total volume of 15 μL, 4 mM ATP, 2 mM MgCl$_2$, 1 mM TCEP, 0.5 mM CoA, 2-$^{14}$C-malonic acid (0.1 μCi), 10 μM MatB, 17 μM PKS, and 50 mM HEPES, pH 8.0. After 2 h of incubation at 25 °C, samples were quenched by adding an equal volume of 1× SDS sample buffer. Following SDS–PAGE analysis with a 4–15% TGX gel (Criterion), the gel was dried for 2 h at 50 °C and exposed on a storage phosphor screen (20 × 25 cm; Molecular Dynamics) for 5 days. Radiolabeled protein was imaged on a Typhoon 9400 phosphorimager (Storage Phosphor mode, 50 μm resolution; Amersham Biosciences).

For in vitro product assays (50 μL) of the PKS protein, 8 μM of PKS was incubated with malonyl-CoA (2 mM) in phosphate buffer (100 mM, pH 7.0) at room temperature for 2 h to generate compound **4**, identified by comparison to a chemical standard. NADPH (2 mM) was added to the assay to generate compounds **5** and **6**, both of which were identified by comparison to chemical standards. Following incubation, the assay mixtures were extracted twice with 100 μL of 99% ethyl acetate/1% acetic acid (v/v). The organic extracts were dried and resuspended in 100 μL of methanol, and 10 μL was injected onto an Agilent Technologies 6120 Quadrupole LC-MS (with DAD) instrument fitted with an Agilent Eclipse Plus C18 column (4.6 × 100 mm). A linear gradient of 2–98% CH$_3$CN (vol/vol) over 40 min in H$_2$O with 0.1% formic acid (vol/vol) at a flow rate of 0.5 mL/min was used. LC-HRMS analysis of the assay extracts was performed on an Agilent Technologies 6520 Accurate-Mass QTOF LC-MS instrument fitted with an Agilent Eclipse Plus C18 column (4.6 × 100 mm) using the same solvent gradient and flow rate described above.

**Data availability.** RNA-seq data generated in this study have been deposited in ArrayExpress (https://www.ebi.ac.uk/arrayexpress/) under accession number E-MTAB-6019. All other relevant data are available from the authors upon request.

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

## Acknowledgements

We thank J. Liu (UC Berkeley) for performing the autoradiography labeling experiment, J. Pelton (UC Berkeley) for helping with NMR spectroscopic analysis, the University of California-Berkeley QB3 Functional Genomics Laboratory and Vincent J. Coates Genomic Sequencing Laboratory for performing the RNA-Seq library preparation and sequencing, the Papoutsakis Lab (University of Delaware) for providing plasmids pKO_mazF and pAN1, and the Tang Lab (UCLA) for providing chemical standards of compounds **4**–**6**. This research was financially supported by the Energy Biosciences Institute, the Pew Scholars Program, the National Institutes of Health (DP2AT009148), and the Chan Zuckerberg Biohub Investigator Program.

## Author contributions

N.A.H. and W.Z. designed the experiments, analyzed the data, and wrote the manuscript. S.J.K. and W.C. performed purification and structure characterization of the polyketides. H.K. assisted with interpretation of NMR spectra of the polyketides. J.S.L. assisted with fermentation and phenotype characterization of *C. acetobutylicum* strains. N.A.H. performed all other experiments.

## Additional information

**Competing interests:** The authors declare no competing financial interests.

