## [Peer Review File · Nature Communications]

Reviewers' comments:

Reviewer #1 (Remarks to the Author):

This is a strong story and paper. It reports on the identification of a novel family of polyketides in this important industrial anaerobe and that the produced polyketides influence the differentiation/sporulation of this organism. The work reported here represents an enormous amount of experiments, but more significantly, a carefully executed strategy, thoughtfully devised and delineated. The writing is exceptionally strong: logical, dense, thorough. Strong figures too that capture the essence and the impact of the data they generated. I should qualify my assessment by stating that I take the XCMS and NMR analyses at face value as I have little expertise to independently verify their data interpretation. My overall sense however is that those analyses were done carefully and expertly given the expertise of the senior author, so I am not concerned about that aspect.

There is only one essential weakness that can be taken care of through a standard revision. Namely that the data used to claim that the polyketides influence solvent production (Fig. 2) are weak and not reliable to support this as a major claim. This is consistent with the fact that they show that there is no transcriptional changes to support the claim. Transcriptional changes are not necessary to change fluxes, but in this and many prokaryotes, this is typically the case for primary metabolites. But this is not an important issue here. Instead, the data that show that the polyketides affect sporulation/differentiation (Fig. 4) are quite convincing and could easily explain the minor effects (Fig. 2) on solvent production. Indeed, if they observed changes in solvent production are robustly different, that could be easily explained by what they state in the Discussion in pp. 14, 15: "This may be explained by the decreased commitment of cells to sporulation in Δ pks (yielding a higher proportion of cells capable of solvent production), as well as the upregulation of cellular machinery related to butanol stress and adaptation as indicated by transcriptomic analysis".

Sporulation/differentiation has been previously shown to affect solvent production beyond the impact of Spo0A. Beyond their ref. 34, there are several other studies with precise KO or KDs of sporulation sigma factors demonstrating altered sporulation and solvent production. E.g. the detailed Genome Biology paper, <http://genomebiology.com/2008/9/7/R114>, showing that sporulation specific sigma factors affecting sporulation and solvent formation and the SigF, sigG and SigE KO papers published in J. Bac in 2011. So, I would downplay the impact on solvent formation and emphasize the robust data on sporulation and granule formation.

A small note: it will help the reader to explain the essence of the iodine assay of Fig. 4b and what do the different colors mean, especially the very dark color of the Δ pks supplemented with clostrienose (3).

Reviewer #2 (Remarks to the Author):

In the manuscript "The industrial anaerobe *Clostridium acetobutylicum* uses polyketides to regulate butanol production and differentiation" Zhang and coworkers report the discovery of a polyketide metabolite of *Clostridium acetobutylicum*. On the basis of mutagenesis and physiological investigations, the authors concluded that the polyketide acts as a signal molecule that influences butanol production and triggers sporulation and granulose accumulation. Despite the good workload put into this paper, the authors overstate some of their findings as some conclusions were not backed up with experiments. For a top journal like Nature Communications one would expect a deeper insight into the potential regulatory role of the compound. There are various major points that needed to be addressed, no matter where this work will be published:

Major points:

Page 8. The authors performed in vitro activity of PKS. It is possible that pyrones are produced as shunt product, but this needs to be confirmed, e.g. with synthetic references.

Page 11. The authors argued that Spo0A, a master regulator of sporulation, was not significantly affected in delta pks strain.

However sigma factors EFG were downregulated as shown in supplementary table 3.

At least, sigF seems to be the first switch for sporulation in *C. acetobutylicum*.

SigE is very likely essential for granulose accumulation in *C. acetobutylicum*.

Spo0A is an upstream regulator of sigF gene, but if the authors do not show any comparable RNA-Seq data of pos. control (less spore forming mutant), one cannot judge on the magnitude of transcription.

It is very likely that sigH is upstream of Spo0A in the *C. acetobutylicum* signal transduction cascade. Some previous reports have shown that Spo0A kinase Cac0903/3319 and Cac0323 could activate Spo0A. On the other hand, Cac0437 has a phosphatase activity against Spo0A. However, only Cac0903 is 2-folds up-regulated in RNA-Seq experiments.

This means Spo0A could be activated (up-regulated), but Spo0A was almost the same

transcription level as wild type. Thus, the RNA-Seq results do not appear to be conclusive or may not be reliable.

Please note that a recent study showed that sigK acts early and late stage of sporulation in *C. acetobutylicum*.(DOI:10.1128/JB.01103-13)

Yet, in the supplementary table 3, sigK gene was down-regulated.

In the above-mentioned report, sigK is strongly upregulated in the middle to late stationary phase. Thus, if RNA-Seq data (26 hours) are reliable, Spo0A is skipped somehow.

A sequential and stage-specific activation of the sporulation-specific sigma factors: sigK-sigH-Spo0A-sigF-sigE-sigG-sigK from some reports.

Page 12. The authors indicated that the colony morphology was different between wild-type and delta pks mutant. This may be an effect of the polyketide (3) as a surfactant. To test this, the authors should carry out an experiment assaying the surfactant activity (ability) of the polyketide (3).

Page 13-14. If clostrienose (3) is a surfactant, it might be not really a signaling molecule. One cannot exclude that a surfactant alters the environment of the bacteria, and the bacteria respond to this by regulating a delay of spore formation.

Page 14. "although none of...molecules.":

It may be a true, but Steiner et al. have reported that the agr KO mutant in *C. acetobutylicum*, which reduced the production of granulose and spore formation, was complemented by the synthetic auto inducer peptide designed from agrD. (see DOI; 10.1128/AEM.06376-11)

How do the authors explain agr quorum-sensing system as a relation to clostrienose (3)?

This one is also related to spore formation, but not to solvenogenesis.

Page 15. The authors summarize clostrienose affects solvent production, but it could be an indirect effect derived from spore formation/granulose accumulation inhibitions.

The results authors showed in this study do not provide sufficient evidence for the function of the polyketide as a signaling molecule.

Structure elucidation:

The authors should show at least the relative stereochemistry by coupling constants and/or NOE correlations.

The authors should provide clear evidence for the sugar moiety of compound 3. The comparison

of NMR data with ppm numbers is not always unambiguous.

Minor comments

Overall, the authors may want to tone down their claims and avoid hyperbole language.

E.g. in Abstract

- What are "interesting biological phenotypes?"
- significant addition... improve traits significant... which P value?
- showcases a novel strategy: this needs to be proven

Page 7. line 124; changes dqf to DQF.

Page 8. line 143-146. Please make a table for a comparison of NMR spectral data of disaccharide.

Supplementary Figure 6.(e) ¹H,¹H-COSY spectrum...of 1, but this spectrum is DQF-COSY spectrum.

Supplementary Figure legends. MeOD could be CD₃OD or MeOH-d₄?

Reviewer #3 (Remarks to the Author):

The manuscript by Herman et al. describes the identification of polyketide metabolites from anaerobic bacteria of the genus *Clostridium*. This is only the second report of polyketides from anaerobes and the first of type I polyketides.

Structure elucidation of the polyketides and interrogation of the PKS activity in vitro are presented. In addition, effects on solvent production, sporulation, granulose accumulation and morphology were studied using a combination of mutagenesis, transcriptome analyses and chemical complementation studies.

The studies were well designed and presented, and the conclusions are sound. The manuscript is well written and a pleasure to read; it should be of interest to readers of Nature Communications. I have only minor comments:

Page 8, lines 155-158: a figure depicting the proposed biosynthesis of compounds 1, 2 and 3 may be useful to a broader audience of Nature Communications readers.

Page 9, line 164: I suggest modifying the following sentence from “...showed a dominant product, the triketide lactone 4, which is a typical...” to “...showed a dominant product with high resolution mass consistent with the triketide lactone 4, which is a typical...”

Page 10, line 167: I suggest adding “according to HRMS analysis” after “pyrone 5 and pentaketide pyrone 6”.

Page 13, line 278: I suggest defining agr, i.e. instead of “well-studied agr quorum sensing system”, “well-studied accessory gene regulator (agr) quorum sensing system”.

Page 15, line 304, Discussion: Would the authors like to discuss the possible influence of precursor supply to the observed increase in solvent production in the pks mutant? If the pks is deleted, acetate/malonate that would be otherwise used for clostrienoic acid and clostrienose biosynthesis is now available and could be diverted to solvent production?

SI, page 40, Supplementary Fig. 9: HRMS spectra have an empty rectangle at the top of the y axis and some of the UV spectra have horizontal double lines going across the spectrum – what are these?

SI, page 42, Supplementary Fig. 11: a shadow of numbers can be seen below the x axis (shifted by one day, i.e. 2 below 1, 3 below 2....). Could the authors please clarify what this is?

Reviewers' comments: (responses in blue)

Reviewer #1 (Remarks to the Author):

This is a strong story and paper. It reports on the identification of a novel family of polyketides in this important industrial anaerobe and that the produced polyketides influence the differentiation/sporulation of this organism. The work reported here represents an enormous amount of experiments, but more significantly, a carefully executed strategy, thoughtfully devised and delineated. The writing is exceptionally strong: logical, dense, thorough. Strong figures too that capture the essence and the impact of the data they generated. I should qualify my assessment by stating that I take the XCMS and NMR analyses at face value as I have little expertise to independently verify their data interpretation. My overall sense however is that those analyses were done carefully and expertly given the expertise of the senior author, so I am not concerned about that aspect.

There is only one essential weakness that can be taken care of through a standard revision. Namely that the data used to claim that the polyketides influence solvent production (Fig. 2) are weak and not reliable to support this as a major claim. This is consistent with the fact that they show that there is no transcriptional changes to support the claim. Transcriptional changes are not necessary to change fluxes, but in this and many prokaryotes, this is typically the case for primary metabolites. But this is not an important issue here. Instead, the data that show that the polyketides affect sporulation/differentiation (Fig. 4) are quite convincing and could easily explain the minor effects (Fig. 2) on solvent production. Indeed, if they observed changes in solvent production are robustly different, that could be easily explained by what they state in the Discussion in pp. 14, 15: "This may be explained by the decreased commitment of cells to sporulation in Δ pks (yielding a higher proportion of cells capable of solvent production), as well as the upregulation of cellular machinery related to butanol stress and adaptation as indicated by transcriptomic analysis". Sporulation/differentiation has been previously shown to affect solvent production beyond the impact of Spo0A. Beyond their ref. 34, there are several other studies with precise KOs or KDs of sporulation sigma factors demonstrating altered sporulation and solvent production. E.g. the detailed Genome Biology paper, <http://genomebiology.com/2008/9/7/R114>, showing that sporulation specific sigma factors affecting sporulation and solvent formation and the SigF, sigG and SigE KO papers published in J. Bac in 2011. So, I would downplay the impact on solvent formation and emphasize the robust data on sporulation and granuloose formation.

We are very thankful for the reviewer's encouraging comments. We agree that a direct influence of polyketide production on solvent production is not supported as strongly compared to our other major conclusions, so we have removed all instances of these claims from the main text (as well as the title). We have also included additional comments to emphasize the robust data on sporulation and granuloose formation.

A small note: it will help the reader to explain the essence of the iodine assay of Fig. 4b and what do the different colors mean, especially the very dark color of the Δ pks supplemented with clostrienose (3).

We have added a more detailed explanation of the iodine assay in the main text to help describe the meaning of **Figure 4b**.

Reviewer #2 (Remarks to the Author):

In the manuscript "The industrial anaerobe *Clostridium acetobutylicum* uses polyketides to regulate butanol production and differentiation" Zhang and coworkers report the discovery of a polyketide metabolite of *Clostridium acetobutylicum*. On the basis of mutagenesis and physiological investigations, the authors concluded that the polyketide acts as a signal molecule that influences butanol production and triggers sporulation and granule accumulation. Despite the good workload put into this paper, the authors overstate some of their findings as some conclusions were not backed up with experiments. For a top journal like *Nature Communications* one would expect a deeper insight into the potential regulatory role of the compound. There are various major points that needed to be addressed, no matter where this work will be published:

Major points:

Page 8. The authors performed *in vitro* activity of PKS. It is possible that pyrones are produced as shunt product, but this needs to be confirmed, e.g. with synthetic references.

We have now analyzed chemical standards of the pyrones provided by Professor Yi Tang from UCLA, and have concluded that compounds 4-6 show the same retention time, high resolution mass spectra, and UV signature as their respective chemical standards. Thus, we are now confident of our structural assignments to compounds 4-6.

Page 11. The authors argued that Spo0A, a master regulator of sporulation, was not significantly affected in Δ *pks* strain. However sigma factors EFG were downregulated as shown in supplementary table 3.

At least, *sigF* seems to be the first switch for sporulation in *C. acetobutylicum*. *SigE* is very likely essential for granule accumulation in *C. acetobutylicum*. Spo0A is an upstream regulator of *sigF* gene, but if the authors do not show any comparable RNA-Seq data of pos. control (less spore forming mutant), one cannot judge on the magnitude of transcription.

The gene in question, CA_P0157 (annotated as a *sigF/sigE/sigG* family sigma factor in supplementary table 3), is not equivalent to the well-studied sigma factors (*sigE* [CA_C1695], *sigF* [CA_C2306], and *sigG* [CA_C1696])^{1,2}. Transcription of these three well-studied factors was not affected by *pks* deletion 26 hours post-inoculation. As stated in the main text, transcription of *spo0A* [CA_C2071] was also unaffected by *pks* deletion at this time point. However, as we stated in the main text, *sigK* [CA_C1689] was downregulated in the *pks* mutant at this time. Please see below for additional discussion on this point.

It is very likely that *sigH* is upstream of Spo0A in the *C. acetobutylicum* signal transduction cascade. Some previous reports have shown that Spo0A kinase Cac0903/3319 and Cac0323

could activate Spo0A. On the other hand, Cac0437 has a phosphatase activity against Spo0A. However, only Cac0903 is 2-folds up-regulated in RNA-Seq experiments. This means Spo0A could be activated (up-regulated), but Spo0A was almost the same transcription level as wild type. Thus, the RNA-Seq results do not appear to be conclusive or may not be reliable.

The orphan histidine kinases CA_C0323, CA_C0903, and CA_C3319 are important for early activation (before 10 hours post-inoculation) of Spo0A in *C. acetobutylicum* ATCC 824³. However, following this early activation, *spo0A* is known to be strongly expressed throughout the remainder of wild-type fermentations, in part, due to upregulation of a number of other regulatory proteins later in the fermentation⁴⁻⁶. Thus, it is uncertain if an upregulation in CA_C0903 in Δpks (relative to wild-type) 26 hours post-inoculation is meaningful for regulation of the already highly expressed Spo0A. Combined with the possibility of post-translational regulation of CA_C0903 and/or Spo0A, and the potential countering effects of the Spo0A dephosphorylase CA_C0437³, we do not believe that upregulation of CA_C0903 in Δpks (relative to wild-type) at this late fermentation time leads to the conclusion that the RNA-Seq results are inconclusive or unreliable. Conclusions that we have made from the RNA-Seq results are based on the expression changes in large numbers of functionally related genes (**Figure 3**), rather than conclusions based on single genes which may or may not be relevant to observed phenotypes.

The validity of our RNA-Seq results is most clearly demonstrated by the observed downregulation of 33 genes related to sporulation in Δpks (relative to wild-type) (**Figure 3 and Supplementary Table 3**), which we confirmed experimentally in both liquid and solid media (**Figure 4 and Supplementary Figure 11**).

Please note that a recent study showed that sigK acts early and late stage of sporulation in *C. acetobutylicum*.(DOI:10.1128/JB.01103-13)

Yet, in the supplementary table 3, sigK gene was down-regulated.

In the above-mentioned report, sigK is strongly upregulated in the middle to late stationary phase. Thus, if RNA-Seq data (26 hours) are reliable, Spo0A is skipped somehow.

A sequential and stage-specific activation of the sporulation-specific sigma factors: sigK-sigH-Spo0A-sigF-sigE-sigG-sigK from some reports.

We thank the reviewer for their suggestion to clarify the potential role of SigK in our work, and have included a paragraph in the main text discussing some of the concepts and hypotheses which we provide here.

It was reported recently that SigK performs two developmentally separated roles in *C. acetobutylicum* ATCC 824, one in early sporulation and one in late sporulation⁷. In early sporulation, SigK is important for upregulation of Spo0A, the master regulator of sporulation and solvent production. For this early role, *sigK* transcription requires upregulation by SigE. Since Spo0A is important for initiating both solventogenesis and sporulation, *sigK* deletion results in low solvent, non-sporulating cultures. In late sporulation, SigK is important for stage IV spore development, including assembly of the spore coat. For this late role, SigK activation is SigG dependent, likely through contributions of the SigG dependent genes *spoIVFB* [CA_C1253] and *spoIVB* [CA_C2072], which are proposed to be required for post-translational processing of pro-SigK to the mature SigK form⁷. This later role of SigK was demonstrated by creating strain $\Delta sigK$ p94Spo0A, which features a *sigK* deletion and plasmid-based overexpression of *spo0A*, thus bypassing the early role of SigK in stimulating Spo0A⁷. Strain $\Delta sigK$ p94Spo0A resulted in cultures with wild-type levels of solvent production, but yielded cells which halted sporulation at stage IV.

In the context of our RNA-Seq results, it appears that *sigK* downregulation in Δpks (at 26 hours post-inoculation) is relevant to the role of *sigK* in late sporulation. This is best illustrated by comparison of Δpks (generated in our study) to $\Delta sigK$ p94Spo0A (generated previously to disrupt only the late role of *sigK*⁷). Solvent production is close to wild-type levels in both strains, suggesting normal activation of Spo0A (associated with the early role of SigK). Sporulation is severely diminished in both strains, by several orders of magnitude in Δpks and complete abolishment in $\Delta sigK$ p94Spo0A. Specifically, sporulation halted at stage IV in $\Delta sigK$ p94Spo0A as evidenced by TEM imaging and comparison to the role of *B. subtilis* SigK in stage IV sporulation⁸. Similarly, many stage IV sporulation genes (including many genes encoding spore coat proteins) were downregulated in Δpks (**Supplementary Table 3**), suggesting that weakened stage IV sporulation was the cause of reduced sporulation rates in Δpks . Furthermore, we also observed downregulation of *spoIVFB* [CA_C1253] and *spoIVB* [CA_C2072] in Δpks (**Supplementary Table 3**), which encode proteins proposed to be required for post-translational processing of pro-SigK to the mature SigK form. Taken together, these results suggest that the *C. acetobutylicum* polyketides are important (but not essential) for stimulating the later role of SigK required for stage IV sporulation.

As noted by the reviewer, our observed downregulation of *sigK* in Δpks (relative to wild-type) is likely not controlled directly by changes in transcription of the well-characterized upstream regulators (*spo0A*, *sigE*, *sigF*, *sigG*), as we did not observe significant changes in transcription of these genes comparing wild-type and Δpks . We propose that the polyketides act to regulate late SigK activity through an unknown mechanism between SigG and SigK, possibly involving contributions in post-translational regulation of SigK from the SigG dependent enzymes SpoIVFB and SpoIVB. While extensive follow up work would be required to validate this hypothesis, this proposed mechanism would fit well into the existing framework of the *Clostridium* sporulation model, as the means by which SigG activates late SigK activity is currently unknown⁴.

Page 12. The authors indicated that the colony morphology was different between wild-type and delta pks mutant. This may be an effect of the polyketide (3) as a surfactant. To test this, the authors should carry out an experiment assaying the surfactant activity (ability) of the polyketide (3).

We thank the reviewer for this thoughtful suggestion. We have now performed two standard assays for surfactant activity—the oil dispersion assay^{9–11} and drop collapse assay^{10,12,13} using pure clostrienose. As shown in the current **Supplementary Table 4**, clostrienose displayed weak surfactant activity at relatively high concentrations (100 μ M), but little to no surfactant activity near physiological concentrations (10 μ M) according to the oil spreading assay. Surfactant activity was not observed using the drop collapse assay at either concentration of clostrienose (10 or 100 μ M), although this is perhaps not surprising given that this assay is known to be less sensitive compared to the oil spreading assay. These results suggest that while clostrienose likely possesses some surfactant activity at high concentrations (approaching mM levels), little or no surfactant activity is observable at concentrations we observed in fermentation culture (< 6 μ M). However, in the context of colony growth on solid media, it is possible that the local concentration of clostrienose along the colony perimeter might be high enough to play a role in surface motility. We have now incorporated these results into **Supplementary Table 4**, and discuss the results in the main text.

Page 13-14. If clostrienose (3) is a surfactant, it might be not really a signaling molecule.

One cannot exclude that a surfactant alters the environment of the bacteria, and the bacteria respond to this by regulating a delay of spore formation.

The presence of clostrienose is actually associated with increased sporulation, not delayed sporulation (**Figure 4**). Currently, we cannot assert the precise regulatory mechanism by which clostrienose affects sporulation in *C. acetobutylicum*, however, our RNA-Seq results as well as previously published studies evaluating the effect of surfactants on gene regulation suggest that a signaling mechanism may be responsible. In the well-studied case of surfactin (widely regarded as a quorum sensing signaling molecule¹⁴⁻¹⁶), this surfactant is known to regulate gene expression (inducing extracellular matrix production) by forming membrane pores leading to potassium ion leakage in *B. subtilis*. The decrease in intracellular potassium is sensed by the membrane histidine kinase KinC, which then activates the regulatory circuit necessary for extracellular matrix production^{14,15}. In another case, *N*-acylhomoserine lactones (AHLs, a well-known class of signaling molecules^{17,18}) natively produced by the bacterium *Rhizobium etli* were shown to act as both biosurfactants and quorum sensing molecules to promote surface colonization¹⁹. These cases indicate that biosurfactants are not restricted to their physiochemical role in decreasing surface tension, and can also control gene expression by acting as signaling molecules in systems like these. Thus for our case, the surfactant activity of clostrienose does not rule out an additional role as a signaling molecule. Furthermore, as discussed in the previous response, little or no surfactant activity was observed with clostrienose at concentrations observed in fermentation culture (< 6 μ M).

We also note that classification as a signaling molecule is also highly dependent on the definition of “signaling molecule” which varies greatly in literature²⁰. For clarity, we define “signaling molecule” here as a biologically-derived secreted small molecule which acts to directly or indirectly regulate gene expression (beyond that related to metabolizing or detoxifying the molecule)²⁰. While this is a rather broad definition, it captures the diverse regulatory strategies harnessed by microbes to communicate and control gene expression (as in the *B. subtilis* case described above). Although significant future work would be required to fully validate clostrienose as a signaling molecule and determine its precise regulatory mechanism in regards to sporulation, we believe this is a reasonable hypothesis, and is presented as such.

We have now incorporated a number of the above discussions into the main text to expand on some of these interesting starting points for future work.

Page 14. "although none of...molecules.":

It may be a true, but Steiner et al. have reported that the agr KO mutant in *C. acetobutylicum*, which reduced the production of granulose and spore formation, was complemented by the synthetic auto inducer peptide designed from agrD. (see DOI; 10.1128/AEM.06376-11)

How do the authors explain agr quorum-sensing system as a relation to clostrienose (3)?

This one is also related to spore formation, but not to solvenogenesis.

While chemical complementation of granulose accumulation and spore formation was demonstrated using a synthetic auto-inducing peptide (AIP) by Steiner *et al.*²¹, this is not equivalent to the extensive purification and characterization required to report the structure a novel bioactive small molecule. However, given that this study is important and relevant to our work, we had cited it in the main text.

In regards to the relationship of the *agr* system and the *pks* system, we cannot confidently state whether these processes are related. Although both systems are important for sporulation and

granulose accumulation, the impact on sporulation rates appear to be different between the two systems. While Steiner *et al.* observed 1-2 order of magnitude decreases in liquid sporulation rates and 1-4 order of magnitude decreases in colony sporulation rates (depending on which of the four *agr* cluster genes was disrupted)²¹, we observed 3-4 order of magnitude decreases in liquid sporulation rates and 2-3 order of magnitude decreases in colony sporulation rates (**Figure 4** and **Supplementary Figure 11**). Thus, the *pks* system appears to be more important for liquid sporulation, while the *agr* system appears to be more important for sporulation on solid media.

Furthermore, we did not observe transcriptional changes in any of the four *agr* cluster genes when comparing Δpks and wild-type at 26 hours post-inoculation, suggesting that these systems may not be directly related. As stated above (and now in the main text), we hypothesize that clostrienose may act as a stimulator of late-state SigK activity. Steiner *et al.* did not suggest a specific mode of action (other than possible regulation of sporulation regulators downstream of Spo0A), so we cannot directly compare our proposed mode of action to that of the *agr* system. Given the fact that *agr* and *pks* mutants did not display reduced solvent production, this suggests (as stated by Steiner *et al.* regarding the *agr* system) that these systems are involved somewhere in the complex regulatory circuitry of sporulation downstream of Spo0A, and likely act as contributors to activating parts of this circuitry given that deletion of *agr* and *pks* genes did not completely abolish sporulation. Given the unaffected transcription of the *agr* cluster in Δpks , the difference in sporulation rates between *agr* and *pks* mutants depending on whether growth is in solid or liquid media, we hypothesize that these systems operate independently within the sporulation regulatory network, although further information on the mode of action of both systems would be required before this can be properly validated.

Page 15. The authors summarize clostrienose affects solvent production, but it could be an indirect effect derived from spore formation/granulose accumulation inhibitions. The results authors showed in this study do not provide sufficient evidence for the function of the polyketide as a signaling molecule.

We agree that the impact of clostrienose production on solvent production is likely an indirect effect of its role in regulating sporulation and granulose accumulation. As we stated in the main text, "This [increased butanol production] may be explained by the decreased commitment of cells to sporulation in Δpks (yielding a higher proportion of cells capable of solvent production), as well as the upregulation of cellular machinery related to butanol stress and adaptation as indicated by transcriptomic analysis."

As stated above in response to Reviewer #1 comments, we agree that a direct influence of polyketide production on solvent production is not supported as strongly compared to our other major conclusions, so we have removed all instances of these claims from the main text (as well as the title).

As described above, we believe hypothesizing that clostrienose acts as a signaling molecule to regulate sporulation and granulose accumulation is reasonable based on our definition of a signaling molecule (a biologically-derived secreted small molecule which acts to directly or indirectly regulate gene expression²⁰). Clostrienose is secreted into and accumulates in the extracellular environment, produced at a very specific growth stage (early stationary phase), appears to stimulate a precise regulatory outcome (suggested as late stage SigK activity), and is able to stimulate sporulation and granulose accumulation when exogenously fed to a *pks* mutant.

Whether or not there is a specific receptor which senses the concentration of clostrienose is currently unknown, but we do not consider this to be a prerequisite for a signaling molecule given

our provided definition and the activity of other well-known signaling molecules which do not rely on receptor proteins to elicit a regulatory response (e.g. surfactin¹⁵). We fully believe that the revised manuscript has provided a deeper insight into the regulatory role of polyketides.

Structure elucidation:

The authors should show at least the relative stereochemistry by coupling constants and/or NOE correlations.

We used $^1J_{\text{CH}}$ values to determine the relative stereochemistry of the pyranoside²². The $^1J_{\text{CH}}$ value for Rha is 176.4 Hz, indicating an alpha stereochemistry^{23,24}. The $^1J_{\text{CH}}$ value for the galactofuranosyl unit is 174.2 Hz, which supports the alpha configuration, but it is known that the difference of $^1J_{\text{CH}}$ values between alpha and beta isomers of aldohexofuranosides are small²⁵. However, the $^3J_{\text{H}_1,\text{H}_2}$ value of 4.8 Hz can be used to determine the alpha configuration²⁴. Additionally, the ^{13}C chemical shift values for the Gal unit suggest that it is alpha-galactofuranose²⁶.

The authors should provide clear evidence for the sugar moiety of compound 3. The comparison of NMR data with ppm numbers is not always unambiguous.

We agree that comparison of ppm numbers of a known sugar moiety could be insufficient to assign the structure of 3. In addition of 1D NMR data, our 2D NMR (**Supplementary Fig. 7**) and HRMS/MS (**Supplementary Fig. 5**) results strongly suggested the presence of a disaccharide moiety.

Minor comments

Overall, the authors may want to tone down their claims and avoid hyperbole language.

E.g. in Abstract

- What are "interesting biological phenotypes?"
- significant addition... improve traits significant... which P value?
- showcases a novel strategy: this needs to be proven

We have toned down our claims to accurately reflect that data we obtained.

Polyketides have been shown to, for example, permit obligate anaerobes to survive aerobic environments²⁷ and inhibit the infiltration of neighboring predatory microbes²⁸, phenotypes we believe are of interest to the broader scientific community.

We have removed one of the uses of the word "significant" in the abstract. The other use of "significant" in the abstract clearly represents the colloquial use of the word, rather than meaning "statistical significance". Given that these polyketides are the second known family of polyketides to be discovered from any anaerobic organism, we consider this work to be significant for the field.

As stated in the main text, “Rather than pursuing a traditional metabolic engineering strategy that focuses on the core metabolic pathway for solvent production, our work showcases an alternative approach by manipulating the secondary metabolism of the organism to improve traits significant for industrial ABE fermentation performance.” Given that rational metabolic engineering of the *C. acetobutylicum* central fermentative pathway has dominated attempts to improve industrial traits in this organism over the last 25 years²⁹, exploring the secondary metabolism of this industrial organism represents a novel strategy.

Page 7. line 124; changes dqf to DQF.

We have now changed this as suggested.

Page 8. line 143-146. Please make a table for a comparison of NMR spectral data of disaccharide.

Here is a comparison table of δ_C and δ_H of **3** and known α -D-galactofuranosyl(1->2)- α -L-rhamnopyranoside moieties isolated from other *Hafnia alvei* strain PCM 1190 (O-specific polysaccharide)³⁰ and HMW-EPS from *B. animalis* subsp. lactis IPLA-R (exopolysaccharide)³¹. We did not include this comparison table since this table is not a typical practice to report NMR spectral data.

No.	3^a		O-specific polysaccharide^b		Exopolysaccharide^c	
	δ_C	δ_H	δ_C	δ_H	δ_C	δ_H
1'	93.80	5.98	100.1	5.04	100.25	5.21
2'	79.17	3.89	79.5	4.35	79.02	4.08
3'	71.33	3.70	70.0	3.91	70.17	3.83
4'	73.58	3.41	72.9	3.53	73.25	3.42
5'	72.52	3.67	69.8	3.77	69.77	3.72
6'	17.94	1.26	17.0	1.31	17.23	1.29
1''	103.51	4.95	102.5	5.15	102.12	5.06
2''	78.46	3.97	76.6	4.16	76.78	4.11
3''	74.52	4.30	73.7	4.30	74.13	4.25
4''	82.61	3.82	81.4	3.89	81.50	3.86
5''	71.51	3.64	70.6	3.80	71.09	3.78
6''	64.34	3.60 3.59	63.4	3.69	61.61	3.68 3.90

^a, NMR was acquired in DMSO-*d*₆ at 298.0 K.

^b, NMR was acquired in D₂O at 328.2 K (¹H) and 343.2 K (¹³C).

^c, NMR was acquired in D₂O at 343.2 K

Supplementary Figure 6.(e) ¹H,¹H-COSY spectrum...of 1, but this spectrum is DQF-COSY spectrum.

We have now changed this as suggested.

Supplementary Figure legends. MeOD could be CD₃OD or MeOH-d₄?

We have now changed this to CD₃OD.

Reviewer #3 (Remarks to the Author):

The manuscript by Herman et al. describes the identification of polyketide metabolites from anaerobic bacteria of the genus *Clostridium*. This is only the second report of polyketides from anaerobes and the first of type I polyketides.

Structure elucidation of the polyketides and interrogation of the PKS activity in vitro are presented. In addition, effects on solvent production, sporulation, granulose accumulation and morphology were studied using a combination of mutagenesis, transcriptome analyses and chemical complementation studies.

The studies were well designed and presented, and the conclusions are sound. The manuscript is well written and a pleasure to read; it should be of interest to readers of *Nature Communications*. I have only minor comments:

We thank the reviewer for their positive feedback, and have addressed the minor comments below.

Page 8, lines 155-158: a figure depicting the proposed biosynthesis of compounds 1, 2 and 3 may be useful to a broader audience of *Nature Communications* readers.

We agree that this may be of interest to readers, so we have included a proposed biosynthetic pathway for compounds 1, 2, and 3 in **Supplementary Figure 12**, and a reference to this in the main text.

Page 9, line 164: I suggest modifying the following sentence from "...showed a dominant product, the triketide lactone 4, which is a typical..." to "...showed a dominant product with high resolution mass consistent with the triketide lactone 4, which is a typical..."

Page 10, line 167: I suggest adding "according to HRMS analysis" after "pyrone 5 and pentaketide pyrone 6".

As discussed in comments for Review #2, we have now analyzed chemical standards of **4-6**, and have concluded that compounds **4-6** show the same retention time, high resolution mass,

and UV signature as their respective chemical standards. Thus, we are confident of our structural assignments to compounds 4-6.

Page 13, line 278: I suggest defining agr, i.e. instead of “well-studied agr quorum sensing system”, “well-studied accessory gene regulator (agr) quorum sensing system”.

We have incorporated this change into the main text.

Page 15, line 304, Discussion: Would the authors like to discuss the possible influence of precursor supply to the observed increase in solvent production in the pks mutant? If the pks is deleted, acetate/malonate that would be otherwise used for clostrienoic acid and clostrienose biosynthesis is now available and could be diverted to solvent production?

While this is an interesting hypothesis, the relatively low levels of clostrienoic acid and clostrienose (< 10 μM) translate to very low levels of consumed acetate/malonate. For example, if all of the carbon atoms from peak levels of clostrienoic acid ($\sim 6 \mu\text{M}$) were instead directed to butanol formation, this would translate to an increase of only $\sim 20 \mu\text{M}$ (or $\sim 1 \text{ mg/L}$) butanol. Thus, we expect changes in solvent production are not due to precursor consumption by polyketide biosynthesis.

SI, page 40, Supplementary Fig. 9: HRMS spectra have an empty rectangle at the top of the y axis and some of the UV spectra have horizontal double lines going across the spectrum – what are these?

We apologize for the figure appearance—this was an unintended error in the PDF conversion, and has been fixed in the current version.

SI, page 42, Supplementary Fig. 11: a shadow of numbers can be seen below the x axis (shifted by one day, i.e. 2 below 1, 3 below 2...). Could the authors please clarify what this is?

Please see the above comment—this error has been fixed as well.

References

1. Tracy, B. P., Jones, S. W. & Papoutsakis, E. T. Inactivation of σE and σG in *Clostridium acetobutylicum* Illuminates Their Roles in Clostridial-Cell-Form Biogenesis, Granulose Synthesis, Solventogenesis, and Spore Morphogenesis. *J. Bacteriol.* **193**, 1414–1426 (2011).
2. Jones, S. W., Tracy, B. P., Gaida, S. M. & Papoutsakis, E. T. Inactivation of σF in *Clostridium acetobutylicum* ATCC 824 Blocks Sporulation Prior to Asymmetric Division and Abolishes σE

- and σ G Protein Expression but Does Not Block Solvent Formation. *J. Bacteriol.* **193**, 2429–2440 (2011).
3. Steiner, E. *et al.* Multiple orphan histidine kinases interact directly with Spo0A to control the initiation of endospore formation in *Clostridium acetobutylicum*. *Mol. Microbiol.* **80**, 641–654 (2011).
 4. Al-Hinai, M. A., Jones, S. W. & Papoutsakis, E. T. The *Clostridium* Sporulation Programs: Diversity and Preservation of Endospore Differentiation. *Microbiol. Mol. Biol. Rev.* **79**, 19–37 (2015).
 5. Jones, S. W. *et al.* The transcriptional program underlying the physiology of clostridial sporulation. *Genome Biol.* **9**, R114 (2008).
 6. Alsaker, K. V. & Papoutsakis, E. T. Transcriptional Program of Early Sporulation and Stationary-Phase Events in *Clostridium acetobutylicum*. *J. Bacteriol.* **187**, 7103–7118 (2005).
 7. Al-Hinai, M. A., Jones, S. W. & Papoutsakis, E. T. σ K of *Clostridium acetobutylicum* Is the First Known Sporulation-Specific Sigma Factor with Two Developmentally Separated Roles, One Early and One Late in Sporulation. *J. Bacteriol.* **196**, 287–299 (2014).
 8. FARQUHAR, R. & YUDKIN, M. D. Phenotypic and Genetic Characterization of Mutations in the *spoIVC* Locus of *Bacillus subtilis*. *Microbiology* **134**, 9–17 (1988).
 9. Morikawa, M., Hirata, Y. & Imanaka, T. A study on the structure–function relationship of lipopeptide biosurfactants. *Biochim. Biophys. Acta BBA - Mol. Cell Biol. Lipids* **1488**, 211–218 (2000).
 10. Płaza, G. A., Zjawiony, I. & Banat, I. M. Use of different methods for detection of thermophilic biosurfactant-producing bacteria from hydrocarbon-contaminated and bioremediated soils. *J. Pet. Sci. Eng.* **50**, 71–77 (2006).
 11. Youssef, N. H. *et al.* Comparison of methods to detect biosurfactant production by diverse microorganisms. *J. Microbiol. Methods* **56**, 339–347 (2004).

12. Bodour, A. A. & Miller-Maier, R. M. Application of a modified drop-collapse technique for surfactant quantitation and screening of biosurfactant-producing microorganisms. *J. Microbiol. Methods* **32**, 273–280 (1998).
13. Jain, D. K., Collins-Thompson, D. L., Lee, H. & Trevors, J. T. A drop-collapsing test for screening surfactant-producing microorganisms. *J. Microbiol. Methods* **13**, 271–279 (1991).
14. López, D. & Kolter, R. Extracellular signals that define distinct and coexisting cell fates in *Bacillus subtilis*. *FEMS Microbiol. Rev.* **34**, 134–149 (2010).
15. López, D., Fischbach, M. A., Chu, F., Losick, R. & Kolter, R. Structurally diverse natural products that cause potassium leakage trigger multicellularity in *Bacillus subtilis*. *Proc. Natl. Acad. Sci.* **106**, 280–285 (2009).
16. Shank, E. A. & Kolter, R. Extracellular signaling and multicellularity in *Bacillus subtilis*. *Curr. Opin. Microbiol.* **14**, 741–747 (2011).
17. Camilli, A. & Bassler, B. L. Bacterial Small-Molecule Signaling Pathways. *Science* **311**, 1113–1116 (2006).
18. Williams, P., Winzer, K., Chan, W. C. & Cámara, M. Look who's talking: communication and quorum sensing in the bacterial world. *Philos. Trans. R. Soc. B Biol. Sci.* **362**, 1119–1134 (2007).
19. Daniels, R. *et al.* Quorum signal molecules as biosurfactants affecting swarming in *Rhizobium etli*. *Proc. Natl. Acad. Sci.* **103**, 14965–14970 (2006).
20. Platt, T. G. & Fuqua, C. What's in a name? The semantics of quorum sensing. *Trends Microbiol.* **18**, 383–387 (2010).
21. Steiner, E., Scott, J., Minton, N. P. & Winzer, K. An agr Quorum Sensing System That Regulates Granulose Formation and Sporulation in *Clostridium acetobutylicum*. *Appl. Environ. Microbiol.* **78**, 1113–1122 (2012).

22. Kasai, R., Okihara, M., Asakawa, J., Mizutani, K. & Tanaka, O. ^{13}C nmr study of α - and β -anomeric pairs of d-mannopyranosides and l-rhamnopyranosides. *Tetrahedron* **35**, 1427–1432 (1979).
23. Mizutani, K., Ohtani, K., Kasai, R., Tanaka, O. & Matsuura, H. Nuclear Magnetic Resonance Study on Glycosyl Esters : Glycosyl Esters of 3-O-Acetyloleanolic Acid and Octanoic Acid. *Chem. Pharm. Bull. (Tokyo)* **33**, 2266–2272 (1985).
24. Ghosh, S. & Misra, A. K. Synthesis of the hexasaccharide repeating unit corresponding to the cell wall lipopolysaccharide of *Azospirillum irakense* KBC1. *Tetrahedron Asymmetry* **21**, 2755–2761 (2010).
25. Cyr, N. & Perlin, A. S. The conformations of furanosides. A ^{13}C nuclear magnetic resonance study. *Can. J. Chem.* **57**, 2504–2511 (1979).
26. Fedonenko, Y. P. *et al.* Structure of the O-polysaccharide of the lipopolysaccharide of *Azospirillum irakense* KBC1. *Carbohydr. Res.* **339**, 1813–1816 (2004).
27. Shabuer, G. *et al.* Plant pathogenic anaerobic bacteria use aromatic polyketides to access aerobic territory. *Science* **350**, 670–674 (2015).
28. Müller, S. *et al.* Bacillaene and Sporulation Protect *Bacillus subtilis* from Predation by *Myxococcus xanthus*. *Appl. Environ. Microbiol.* **80**, 5603–5610 (2014).
29. Lütke-Eversloh, T. Application of new metabolic engineering tools for *Clostridium acetobutylicum*. *Appl. Microbiol. Biotechnol.* **98**, 5823–5837 (2014).
30. Petersson, C., Jachymek, W., Kenne, L., Niedziela, T. & Lugowski, C. Structural studies of the O-specific chain of *Hafnia alvei* strain PCM 1190 lipopolysaccharide. *Carbohydr. Res.* **298**, 219–227 (1997).
31. Leivers, S. *et al.* Structure of the high molecular weight exopolysaccharide produced by *Bifidobacterium animalis* subsp. *lactis* IPLA-R1 and sequence analysis of its putative eps cluster. *Carbohydr. Res.* **346**, 2710–2717 (2011).

Reviewers' Comments:

Reviewer #1 (Remarks to the Author):

I am happy with how the authors addressed my concerns, and the overall quality of the revision.

I still somehow felt that the response to the comments by Rev. #2 was not integrated into the revision, and some of that response should. E.g. the relationship between SigK and CA_P0157 makes sense since it was shown that CA_P0157 is a very late sporulation sigma factor (Genome Biology Cac paper) and would make sense to be controlled by SigK. But also the discussion re the role of other sporulation sigma factors on granulose and spore formation as has been elucidate in the literature for this organism. As an aside, Rev. #2 wrote: "SigE is very likely essential for granulose accumulation in *C. acetobutylicum*". Not very likely. It is essential (doi:10.1128/JB.01380-10).

I will leave this up to the editor and Rev. #2 to decide.

Reviewer #2 (Remarks to the Author):

The authors have done a good job addressing (most of) the reviewers' comments.

Reviewer #3 (Remarks to the Author):

The authors have satisfactorily addressed reviewers' comments.

Reviewers' comments: (responses in blue)

Reviewer #1 (Remarks to the Author):

I am happy with how the authors addressed my concerns, and the overall quality of the revision.

I still somehow felt that the response to the comments by Rev. #2 was not integrated into the revision, and some of that response should. E.g. the relationship between SigK and CA_P0157 makes sense since it was shown that CA_P0157 is a very late sporulation sigma factor (Genome Biology Cac paper) and would make sense to be controlled by SigK. But also the discussion re the role of other sporulation sigma factors on granule and spore formation as has been elucidated in the literature for this organism. As an aside, Rev. #2 wrote: "SigE is very likely essential for granule accumulation in *C. acetobutylicum*". Not very likely. It is essential (doi:10.1128/JB.01380-10).

I will leave this up to the editor and Rev. #2 to decide.

We have now included a brief discussion of the potential relationship between SigK and CA_P0157 in the main text, which we agree makes sense given the published expression profile of CA_P0157¹.

We agree that recent work elucidating the role of other sigma factors which regulate sporulation/granule accumulation in *C. acetobutylicum* is important for contextualizing our results and discussion of SigK, so we have added citations in the main text which detail the roles of SigF², SigG/SigE³, SpoIIE⁴, as well as a comprehensive review of the topic⁵. Given that *sigK* was the only well-characterized sporulation-related regulator with significantly altered gene expression in our study, we have chosen to focus our detailed discussion (in the main text) on SigK and closely related regulators (namely, Spo0A, SigE, and SigG).

Reviewer #2 (Remarks to the Author):

The authors have done a good job addressing (most of) the reviewers' comments.

We thank the reviewer for all of their suggestions and comments.

Reviewer #3 (Remarks to the Author):

The authors have satisfactorily addressed reviewers' comments.

We thank the reviewer for their feedback and suggestions.

References:

1. Jones, S. W. *et al.* The transcriptional program underlying the physiology of clostridial sporulation. *Genome Biol.* **9**, R114 (2008).
2. Jones, S. W., Tracy, B. P., Gaida, S. M. & Papoutsakis, E. T. Inactivation of σ F in *Clostridium acetobutylicum* ATCC 824 Blocks Sporulation Prior to Asymmetric Division and Abolishes σ E and σ G Protein Expression but Does Not Block Solvent Formation. *J. Bacteriol.* **193**, 2429–2440 (2011).
3. Tracy, B. P., Jones, S. W. & Papoutsakis, E. T. Inactivation of σ E and σ G in *Clostridium acetobutylicum* Illuminates Their Roles in Clostridial-Cell-Form Biogenesis, Granule Synthesis, Solventogenesis, and Spore Morphogenesis. *J. Bacteriol.* **193**, 1414–1426 (2011).
4. Bi, C., Jones, S. W., Hess, D. R., Tracy, B. P. & Papoutsakis, E. T. SpoIIE Is Necessary for Asymmetric Division, Sporulation, and Expression of σ F, σ E, and σ G but Does Not Control Solvent Production in *Clostridium acetobutylicum* ATCC 824. *J. Bacteriol.* **193**, 5130–5137 (2011).
5. Al-Hinai, M. A., Jones, S. W. & Papoutsakis, E. T. The *Clostridium* Sporulation Programs: Diversity and Preservation of Endospore Differentiation. *Microbiol. Mol. Biol. Rev.* **79**, 19–37 (2015).